# Expanding the genetic architecture of nicotine dependence and its shared genetics with multiple traits

Bryan C. Quach et al.[#]

Cigarette smoking is the leading cause of preventable morbidity and mortality. Genetic variation contributes to initiation, regular smoking, nicotine dependence, and cessation. We present a Fagerström Test for Nicotine Dependence (FTND)-based genome-wide association study in 58,000 European or African ancestry smokers. We observe five genome-wide significant loci, including previously unreported loci *MAGI2/GNAI1* (rs2714700) and *TENM2* (rs1862416), and extend loci reported for other smoking traits to nicotine dependence. Using the heaviness of smoking index from UK Biobank ($N = 33{,}791$), rs2714700 is consistently associated; rs1862416 is not associated, likely reflecting nicotine dependence features not captured by the heaviness of smoking index. Both variants influence nearby gene expression (rs2714700/*MAGI2-AS3* in hippocampus; rs1862416/*TENM2* in lung), and expression of genes spanning nicotine dependence-associated variants is enriched in cerebellum. Nicotine dependence (SNP-based heritability = 8.6%) is genetically correlated with 18 other smoking traits ($r_g = 0.40$–1.09) and co-morbidities. Our results highlight nicotine dependence-specific loci, emphasizing the FTND as a composite phenotype that expands genetic knowledge of smoking.

[#]A list of authors and their affiliations appears at the end of the paper.

Cigarette smoking remains the leading cause of preventable death worldwide[1], despite the well-known adverse health effects. Smoking causes more than 7 million deaths annually from a multitude of diseases including cancer, chronic obstructive pulmonary disease (COPD), and heart disease[1,2]. Cigarette smoking is a multi-stage process consisting of initiation, regular smoking, nicotine dependence (ND), and cessation. Each step has a strong genetic component (for example, twin-based heritability estimates up to 70% for the transition from regular smoking to ND[3,4]) and partial overlaps are expected among the sets of sequence variants correlating with the different stages[3], as evidenced by findings of the GWAS and Sequencing Consortium of Alcohol and Nicotine use (GSCAN) with sample sizes up to 1.2 million individuals[5]. GSCAN identified 298 genome-wide significant loci associated with initiation (ever vs. never smoking), age at initiation, cigarettes per day (CPD), and/or cessation (current vs. former smoking); 259 of the loci harbored significant associations with initiation[5].

In comparison to other stages of smoking, known loci for ND are limited. Only six reproducible, genome-wide significant loci have been identified: *CHRNB3-CHRNA6* (chr8p11), *DBH* (chr9q34), *CHRNA5-CHRNA3-CHRNB4* (chr15q25), *DNMT3B* and *NOL4L* (chr20q11), and *CHRNA4* (chr20q13)[6]. A more complete understanding of the genetics underlying ND is needed, as it could help to predict the likelihood of quitting smoking, withdrawal severity, response to treatment, and health-related consequences[7–10]. The Fagerström Test for ND (FTND), also called the Fagerström Test for Cigarette Dependence[11], provides a composite phenotype that captures multiple behavioral and psychological features of ND among smokers[12]. While CPD is associated with key markers of ND, such as cessation likelihood[13], the FTND conveys additional valuable information by including 5 items in addition to CPD. FTND is meaningfully associated with tobacco use diagnostic criteria from the Diagnostic and Statistical Manual of Mental Disorders[14,15] and is more highly associated with withdrawal severity than is CPD[7]. Its validity may be due to the inclusion of the time-to-first-cigarette in the morning (TTFC) item, which appears to be especially strongly associated with relapse likelihood[16–18] and may be an especially informative measure of heritability of ND[19]. Thus, the FTND provides somewhat different information than CPD alone and has been relatively understudied from a genetic perspective because of its more limited availability across datasets.

The FTND score, based on totaling responses to the 6 items that constitute the FTND, ranges from 0 (no dependence) to 10 (highest dependence level)[12,20]. In the present study, we categorize FTND scores as mild (scores 0–3), moderate (scores 4–6), or severe (scores 7–10), as done before in studies comprising our Nicotine Dependence GenOmics (iNDiGO) Consortium[21,22]. We expand upon our prior analyses and report findings from the largest GWAS meta-analysis for ND ($N = 58,000$; 46,213 European [EUR] ancestry and 11,787 African American [AA] participants from 23 studies). Our findings highlight two genetic loci with previously unreported associations with cigarette smoking, genetic correlations between ND and 18 other phenotypes, and enrichment of ND heritability with genes expressed in cerebellum. By testing GSCAN-identified loci[5], we report loci whose associations with other smoking outcomes, such as CPD, extend to ND. Our findings support a complex polygenetic architecture of ND, with neurobiological indications, including loci shared across smoking traits and ND-specific loci.

## Results

**GWAS meta-analysis finds two novel SNP associations with ND.** Our cross-ancestry ND GWAS meta-analysis ($\lambda = 1.035$, Supplementary Fig. 1A) identified five genome-wide significant loci (Fig. 1). Associations of the lead SNPs (single nucleotide polymorphisms) from each of these five loci are shown in Table 1.

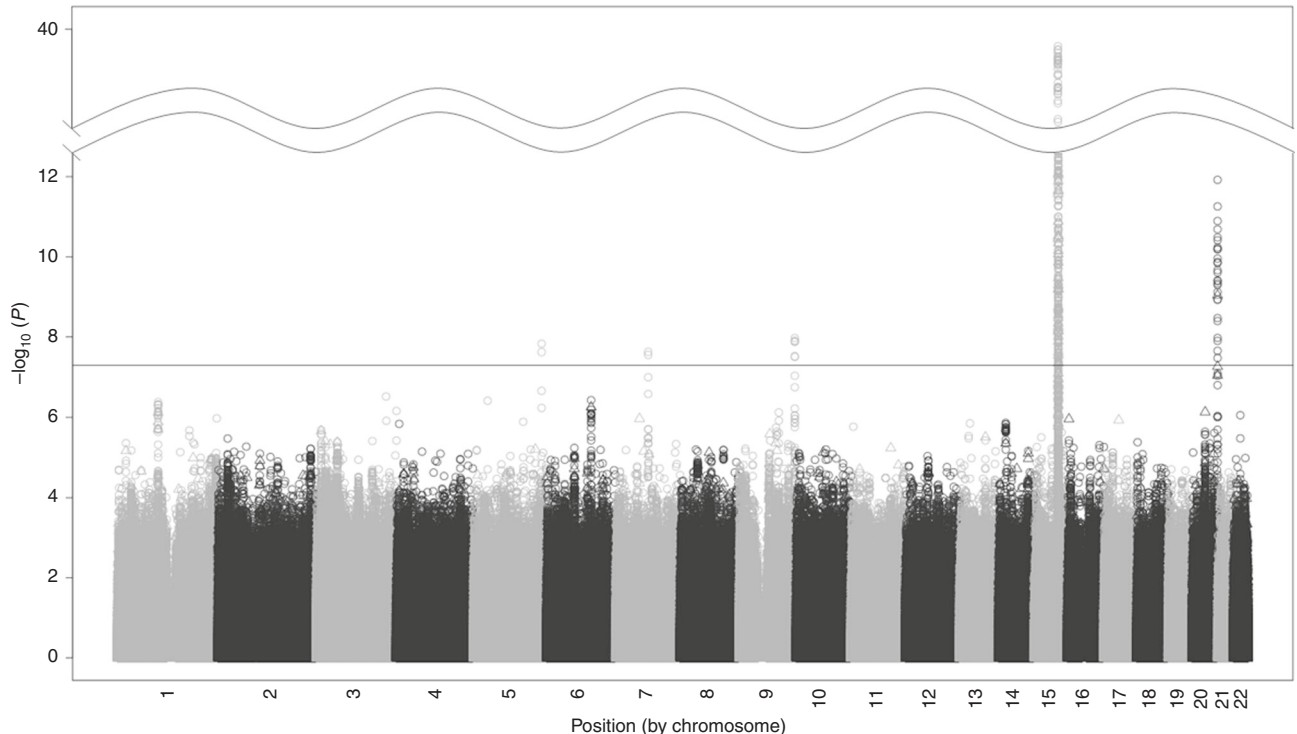

**Fig. 1 Cross-ancestry nicotine dependence genome-wide association meta-analysis results.** This study comprises 23 iNDiGO studies with total $N = 58,000$ biologically independent samples from European and African American ancestry ever smokers. The $-\log_{10}$ meta-analysis *p*-values of single nucleotide polymorphisms (SNPs; depicted as circles) and insertions/deletions (indels; depicted as triangles) are plotted by chromosomal position. Five loci surpassed the genome-wide statistical significance threshold ($P < 5 \times 10^{-8}$, as marked by the solid horizontal black line).

**Table 1 Lead single nucleotide polymorphism (SNP) associations with nicotine dependence for five genome-wide significant loci in the Nicotine Dependence GenOmics (iNDiGO) consortium.**

| SNP (effect allele) | Chr:position (NCBI build 37) | Gene/closest genes | Effect allele frequencies[a] | European ancestry-specific β (SE), P | African American-specific β (SE), P | Cross-ancestry β (SE), P |
|---|---|---|---|---|---|---|
| rs1862416 (T) | 5:167,394,595 | TENM2 | 0.88; 0.94 | 0.037 (0.0074), $5.4 \times 10^{-7}$ | 0.049 (0.0066), $6.6 \times 10^{-3}$ | 0.039 (0.0068), $1.5 \times 10^{-8}$ |
| rs2714700 (T) | 7:79,367,667 | MAGI2/GNAI1 | 0.47; 0.72 | −0.022 (0.0045), $1.2 \times 10^{-6}$ | −0.026 (0.0094), $5.5 \times 10^{-3}$ | −0.023 (0.0041), $2.3 \times 10^{-8}$ |
| rs13284520 (A) | 9:136,502,572 | DBH | 0.83; 0.56 | 0.028 (0.0059), $1.7 \times 10^{-6}$ | 0.029 (0.0092), $1.7 \times 10^{-3}$ | 0.029 (0.0050), $1.1 \times 10^{-8}$ |
| rs16969968 (A) | 15:78,882,925 | CHRNA5 | 0.37; 0.02 | 0.061 (0.0047), $4.9 \times 10^{-38}$ | 0.049 (0.018), $7.1 \times 10^{-3}$ | 0.060 (0.0046), $1.6 \times 10^{-39}$ |
| rs151176846 (T) | 20:61,997,500 | CHRNA4 | 0.92; 1.00 | −0.067 (0.0094), $1.2 \times 10^{-12}$ | NA | −0.067 (0.0094), $1.2 \times 10^{-12}$ |

Cross-ancestry meta-analysis results (total N = 58,000 biologically independent samples) are presented, along with ancestry-specific association results (Ns = 46,213 and 11,787 biologically independent samples from European ancestry and African American individuals, respectively). β values correspond to direction of association for the effect alleles, with standard errors (SEs) shown.
NA not available (due to monomorphism for rs151176846 among African Americans); NCBI National Center for Biotechnology Information.
[a]Frequencies correspond to 1000G European and African superpopulation reference panels, respectively.

All genome-wide significant SNP/indel associations from the cross-ancestry meta-analysis are provided in Supplementary Data 1.

Three of the genome-wide significant loci have known associations with ND from our prior GWAS and others[6]: chr15q25[21–23] (smallest $P = 1.6 \times 10^{-39}$ for rs16969968, a well-established functional missense [D398N] CHRNA5 SNP[24]) chr20q13[21] (smallest $P = 1.2 \times 10^{-12}$ for rs151176846, an intronic CHRNA4 SNP), and chr9q34[22] (smallest $P = 1.1 \times 10^{-8}$ for rs13284520, an intronic DBH SNP). In the EUR-specific GWAS meta-analysis, the loci spanning nicotinic acetylcholine receptor genes (CHRNA5-A3-B4 and CHRNA4), but no novel loci, were identified at genome-wide significance ($\lambda = 1.036$, Supplementary Figs. 1B and 2A). No genome-wide significant loci were found in the GWAS meta-analysis among AAs ($\lambda = 1.032$, Supplementary Figs. 1C and 2B).

Two genome-wide significant loci from the cross-ancestry meta-analysis represent novel associations with ND. On chr7q21, the most significant SNP ($P = 2.3 \times 10^{-9}$) was rs2714700, a SNP between the MAGI2 and GNAI1 genes (Supplementary Fig. 3A–B). The most significant SNP on chr5q34, rs1862416 ($P = 1.5 \times 10^{-8}$), sits within an intron for TENM2 (Supplementary Fig. 3C–D). Both SNPs imputed well: sample size-weighted mean estimated $r^2$ values were 0.97 for rs2714700 and 0.92 for rs1862416. Further, both SNPs were common, and their associations with ND were observed across EURs and AAs (Table 1) and were largely consistent across studies (Supplementary Fig. 4A–B): rs2714700-T being associated with reduced risk (meta-analysis odds ratio [OR] and 95% confidence interval [CI] = 0.96 [0.94–0.97]) and rs1862416-T being associated with increased risk (meta-analysis OR [95% CI] = 1.08 [1.05–1.11]) for severe vs. mild ND. These comparisons of dissimilar categories were derived from the GWAS regression coefficients (i.e., OR = exp[2 × β] for severe vs. mild ND, with OR > 1 corresponding to an increased risk of severe ND) to contextualize the magnitude of the observed effect sizes. Neither SNP showed evidence for heterogeneity, based on the $I^2$ index[25], across studies ($P = 0.83$ for rs2714700 and 0.75 for rs1862416). Leave-one-study-out analyses (Supplementary Table 1) revealed some variability in p-values ($P = 3.1 \times 10^{-7}$–$7.4 \times 10^{-9}$ for rs2714700 and $P = 5.6 \times 10^{-9}$–$3.9 \times 10^{-6}$ for rs1862416), likely due to fluctuating statistical power given the significant correlation between N (sample size) and p-value across iterations: $r = -0.65$, $P = 8.6 \times 10^{-5}$. However, there was little variation in the effect sizes (range of β values corresponding to the OR for severe vs. mild ND = 0.95 – 0.96 for rs2714700-T and 1.07 – 1.08 for rs1862416-T).

We compared the novel ND-associated SNPs with results reported for other smoking traits by GSCAN[5]. European ancestry participants from 8 iNDiGO studies were included in GSCAN (Supplementary Table 2). Both the MAGI2/GNAI1 SNP rs2714700 and the TENM2 SNP rs1862416 were nominally associated at $P < 0.05$ with ever vs. never smoking and rs2714700 with CPD in consistent directions with ND; neither SNP was associated with age at initiation or smoking cessation (Supplementary Table 3). Other SNPs near rs1862416 were associated at genome-wide significance with ever vs. never smoking in GSCAN (Supplementary Table 4).

For replication in an independent sample, we analyzed the two novel SNPs (rs2714700 and rs1862416) for association with the heaviness of smoking index (HSI) in the UK Biobank. Results are shown in Supplementary Table 5. HSI is based on two items (CPD and TTFC) of the 6 items that constitute the FTND; the HSI and full-scale FTND are highly correlated (e.g., $r = 0.7$ among nondaily smokers and 0.9 among daily smokers)[26]. The MAGI2/GNAI1 SNP, rs2714700, was associated with HSI at $P = 0.014$, which surpassed Bonferroni correction for two SNP tests, and meta-analysis of iNDiGO studies with UK Biobank

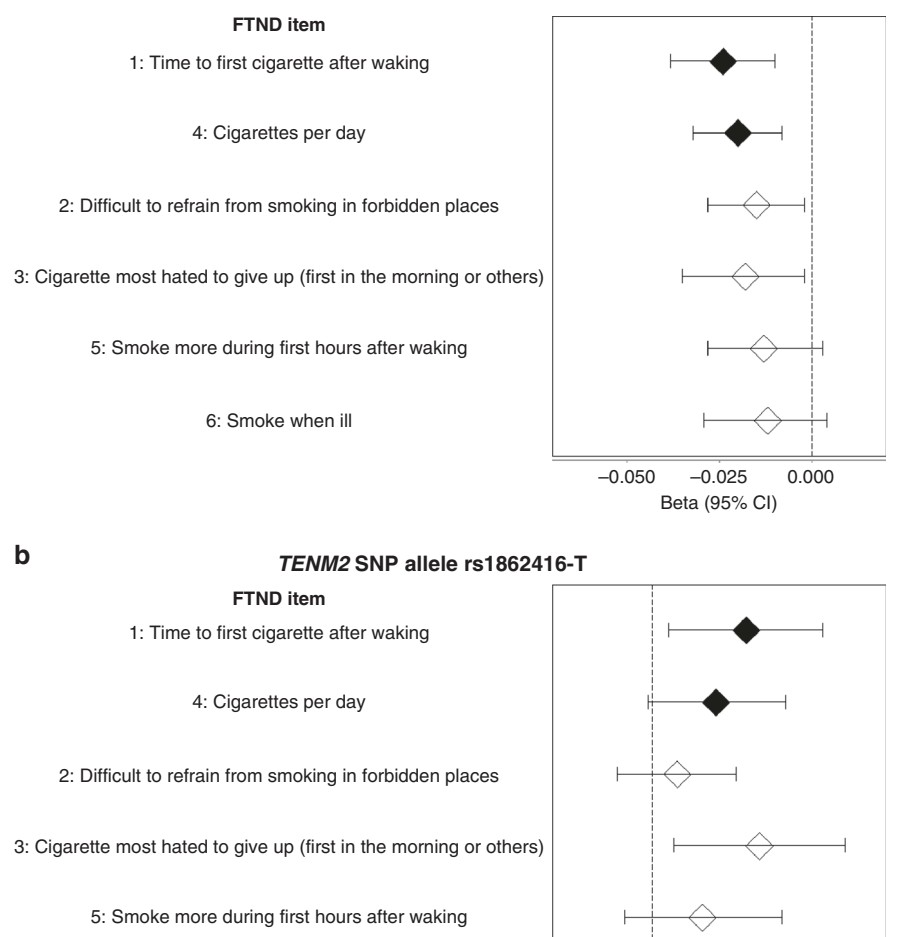

**Fig. 2 Single nucleotide polymorphism (SNP) associations with specific Fagerström Test for Nicotine Dependence (FTND) items.** Beta values and corresponding 95% confidence intervals (CIs) were taken from cross-ancestry meta-analysis of **a** rs2714700 and **b** rs1862416 using linear regression models for categorical FTND item responses (1 and 4, closed diamonds) or logistic regression models for binary FTND item responses (2, 3, 5, and 6, open diamonds) across the iNDiGO studies (N up to 47,569 biologically independent samples with complete FTND data contributing to the specific item analyses). Diamonds indicate the beta values, and error bars correspond to the 95% CI estimates for the beta values.

(total $N = 91,791$) supported rs2714700-T being associated with milder ND ($P = 7.7 \times 10^{-9}$). The *TENM2* SNP, rs1862416, was not associated with HSI in the UK Biobank ($P = 0.39$).

To determine whether the novel genome-wide associations were driven by specific FTND items or shared across items, we returned to the iNDiGO studies, tested SNP associations with each specific FTND item, and combined results via cross-ancestry meta-analyses. For rs2714700, we observed the lowest *p*-values for the two items that constitute the HSI (Fig. 2a): TTFC ($P = 5.3 \times 10^{-4}$) and CPD ($P = 1.1 \times 10^{-3}$). Rs2714700 was also associated at $P < 0.05$ with difficult in refraining from smoking in forbidden places ($P = 0.025$) and the cigarette most hated to give up ($P = 0.030$). Rs1862416 was associated with TTFC ($P = 0.018$) and two items that are not captured by the HSI: the cigarette most hated to give up ($P = 0.015$) and smoking when ill ($P = 0.023$) (Fig. 2b).

**GWAS findings for other smoking traits extend to ND.** We assessed whether genome-wide significant SNPs identified for

smoking traits in GSCAN extended to ND using results from the cross-ancestry GWAS meta-analysis. We focused on the 55 genome-wide significant SNPs from 40 loci associated with CPD, given that it displayed the best genetic correlation with ND (Fig. 3). After applying Bonferroni correction for the 53 SNPs that were available in our meta-analysis ($P < 9.4 \times 10^{-4}$), 17 SNPs had a statistically significant and directionally consistent association with ND (Table 2). These SNPs span six loci reported at genome-wide or nominal significance in prior GWAS of ND (*CHRNA5-A3-B4* [chr15], *CHRNA4* [chr20], *DBH* [chr9], *CHRNB3* [chr8], *CYP2A6* [chr19], and *NOL4L* [near *DNMT3B*, chr20])[6] and three loci not reported in prior ND GWAS—*DRD2* (chr11), *C16orf97* (chr16), and *CHRNB2* (chr1).

**Gene-based association analyses highlight known genetic loci.** Using Hi-C coupled multi-marker analysis of genomic annotation (H-MAGMA)[27] on the EUR-specific GWAS meta-analysis results from iNDiGO, 11 genes when using fetal brain tissue and

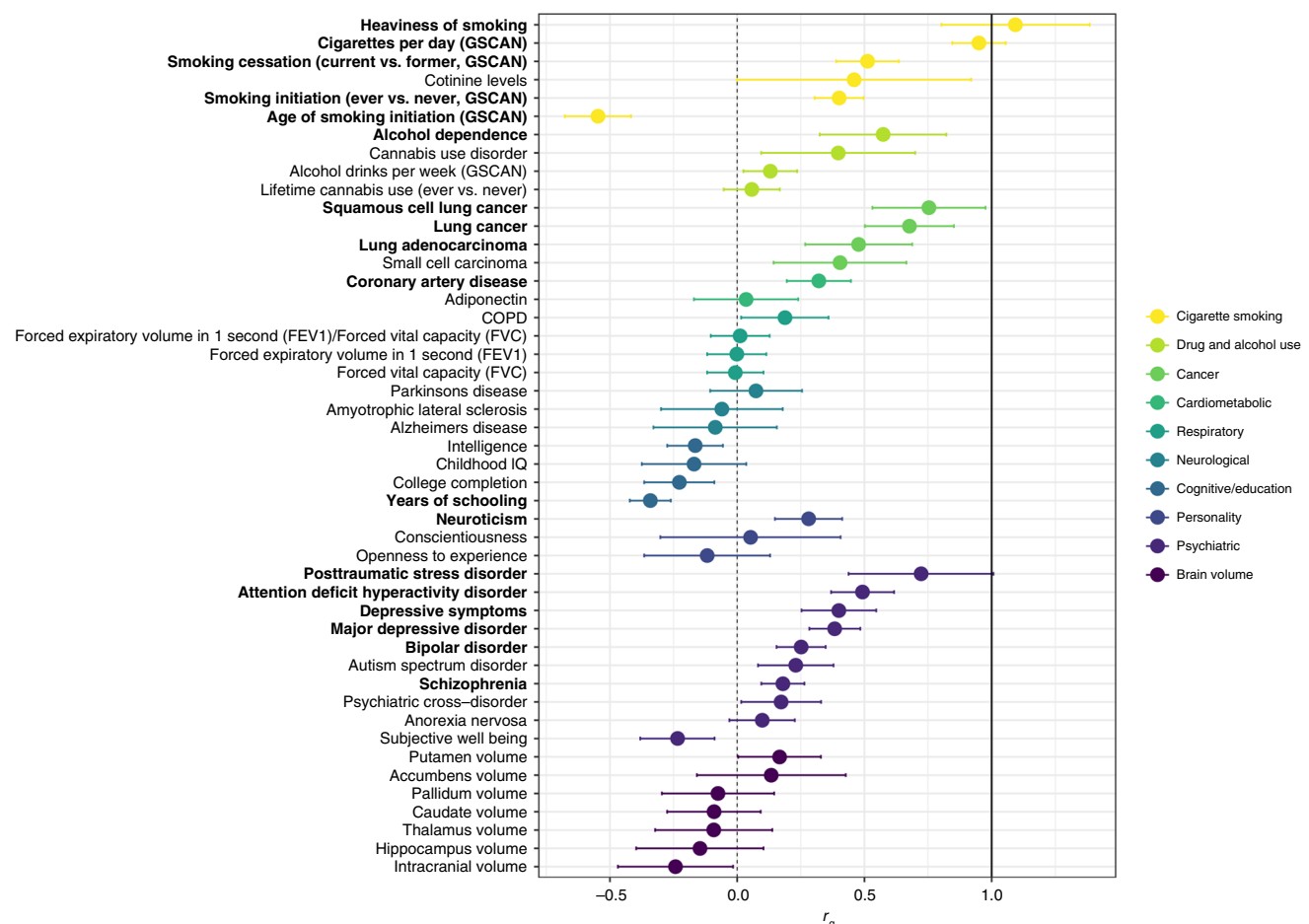

**Fig. 3 Genetic correlations of nicotine dependence (ND) with 47 other phenotypes.** Correlations were calculated using linkage disequilibrium (LD) score regression with the iNDiGO European ancestry-specific GWAS meta-analysis results for ND ($N = 46,213$ biologically independent samples), compared with results made available via LD Hub or study investigators (see Supplementary Table 7 for original references). Phenotypes were grouped by disease/trait or measurement category, as indicated by different colorings. Dots indicate the mean values for genetic correlation ($r_g$); error bars show the 95% confidence intervals; the dashed vertical black line corresponds to $r_g = 0$ (no correlation with ND), and the solid vertical black line corresponds to $r_g = 0$ (complete correlation with ND). Phenotypes with significant correlation with ND are bolded (1 degree of freedom Chi-square test; Bonferroni adjusted $p$-value <0.05 after accounting for 47 independent tests). Exact $p$-values are provided in Supplementary Table 7.

13 genes when using adult brain tissue were associated with ND at $P < 2.7 \times 10^{-6}$, based on correction for testing 18,655 protein coding genes. See Supplementary Data 2 and 3 for the genome-wide H-MAGMA results for fetal and adult tissues, respectively. Of the 16 unique genes identified, 10 genes in three known loci were associated with HSI in the UK Biobank at $P < 0.0031$, based on correction for testing 16 genes (Supplementary Table 6): the *ACSBG1-WDR61-IREB2-HYKK-PSMA4-CHRNA5-CHRNA3-CHRNAB4-ADAMTS7-MORF4L1* gene cluster on chr. 15q25, *CHRNA4* on chr. 20q13, and the *ADAMTSL2* and *DBH* genes in close proximity on chr. 9q34. Two novel genes on distinct chromosomes were identified in iNDiGO (*AFG1L* on chr. 6q21 and *AK2* on chr. 1p35), but their associations were not corroborated in UK Biobank.

We also applied Summary-MultiXcan (S-MultiXcan)[28] to the EUR-specific GWAS meta-analysis results from iNDiGO and found significant associations for two chromosome 15q25 genes (*PSMA4* and *CHRNA5*), when considering *cis*-expression quantitative trait loci (*cis*-eQTL) evidence from either the multi-tissue or single best tissue (substantia nigra). See Supplementary Data 4 for the genome-wide S-MultiXcan results. Both genes were also associated with HSI in UK Biobank from multi-tissue ($P = 2.4 \times 10^{-8}$ for *PSMA4* and $1.3 \times 10^{-6}$ for *CHRNA5*) or single best tissue

($P = 9.6 \times 10^{-14}$ for *PSMA4* and $4.6 \times 10^{-8}$ for *CHRNA5*, both in substantia nigra).

**ND is genetically correlated with 18 other phenotypes.** We estimated the heritability explained by common SNPs of ND at $h_g^2$ (standard error) $= 0.086$ (0.012), using LDSC[29] and the EUR-specific GWAS meta-analysis results. We also found statistically significant genetic correlations of ND with 18 phenotypes (Bonferroni-corrected $P < 0.0011$; Fig. 3 and Supplementary Table 7). Positive correlations indicate that the genetic predisposition to higher ND risk was correlated with genetic risks for other smoking traits[5] (smallest $P = 3.1 \times 10^{-70}$ for higher CPD [$r_g = 0.95$], followed by $P = 3.4 \times 10^{-16}$ for current smoking [$r_g = 0.51$], $P = 3.2 \times 10^{-16}$ for ever smoking [$r_g = 0.40$], and $P = 1.8 \times 10^{-13}$ for HSI [highest $r_g$ at >1]). We repeated LDSC, after removing all chr15q25 variants between 78.5 and 79.5 megabases (MB) and found only negligible differences in these correlations ($r_g = 0.94$ for higher CPD, $r_g = 0.51$ for current smoking, and $r_g = 0.42$ for ever smoking). Beyond the smoking traits, with all SNPs included, higher ND was genetically correlated with higher risks of alcohol dependence[30], neuroticism[31], psychiatric diseases (attention deficit hyperactivity disorder[32], bipolar disorder[33], major depressive

**Table 2 Single nucleotide polymorphisms (SNPs) identified as genome-wide significant for cigarettes per day (CPD) by the GWAS and Sequencing Consortium of Alcohol and Nicotine use (GSCAN) consortium and associated with nicotine dependence (ND) in the Nicotine Dependence GenOmics (iNDiGO) consortium.**

| SNP (effect allele) | Chr:position (NCBI build 37) | Gene/nearest gene(s) | GSCAN $\beta$ (SE), $P$ | iNDiGO $\beta$ (SE), $P$ |
|---|---|---|---|---|
| rs7125588[a] (G) | 11:113,436,072 | DRD2 / TMPRSS5 | −0.014 (0.0020), $6.5 \times 10^{-12}$ | −0.016 (0.0042), $1.8 \times 10^{-4}$ |
| rs1592485[a] (A) | 16:52,093,549 | C16orf97 | −0.013 (0.0021), $1.1 \times 10^{-10}$ | −0.015 (0.0043), $4.5 \times 10^{-4}$ |
| rs2072659[a] (G) | 1:154,548,521 | CHRNB2 | −0.025 (0.0038), $2.5 \times 10^{-13}$ | −0.026 (0.0078), $8.4 \times 10^{-4}$ |
| rs146009840[b] (T) | 15:78,906,177 | CHRNA3 | 0.030 (0.0036), $2.0 \times 10^{-17}$ | 0.060 (0.0046), $2.6 \times 10^{-39}$ |
| rs72740955[b] (T) | 15:78,849,779 | PSMA4 / CHRNA5 | 0.040 (0.0033), $2.4 \times 10^{-34}$ | 0.058 (0.0045), $1.5 \times 10^{-38}$ |
| rs10519203[b] (A) | 15:78,814,046 | HYKK | −0.075 (0.0021), $3.1 \times 10^{-286}$ | −0.050 (0.0042), $7.7 \times 10^{-32}$ |
| rs8040868[b] (C) | 15:78,911,181 | CHRNA3 | 0.022 (0.0034), $1.8 \times 10^{-10}$ | 0.044 (0.0041), $7.3 \times 10^{-27}$ |
| rs12438181[b] (A) | 15:78,812,098 | HYKK | −0.023 (0.0037), $5.0 \times 10^{-10}$ | −0.039 (0.0049), $2.6 \times 10^{-15}$ |
| rs3743063[b] (C) | 15:79,065,171 | ADAMTS7 | −0.023 (0.0035), $1.5 \times 10^{-11}$ | −0.030 (0.0042), $6.8 \times 10^{-13}$ |
| rs28681284[b] (T) | 15:78,908,565 | CHRNA3 | −0.049 (0.0030), $2.1 \times 10^{-58}$ | −0.035 (0.0051), $1.1 \times 10^{-11}$ |
| rs2273500[b] (C) | 20:61,986,949 | CHRNA4 | 0.031 (0.0029), $3.5 \times 10^{-26}$ | 0.034 (0.0058), $4.0 \times 10^{-9}$ |
| rs3025383[b] (C) | 9:136,502,369 | DBH | −0.026 (0.0026), $9.8 \times 10^{-24}$ | −0.025 (0.0049), $1.8 \times 10^{-7}$ |
| rs28438420[b] (T) | 15:78,836,288 | PSMA4 | 0.020 (0.0028), $1.3 \times 10^{-12}$ | 0.020 (0.0041), $7.9 \times 10^{-7}$ |
| rs75596189[b] (T) | 9:136,468,701 | FAM163B / DBH | 0.035 (0.0037), $1.8 \times 10^{-20}$ | 0.030 (0.0066), $8.1 \times 10^{-6}$ |
| rs4236926[b] (G) | 8:42,578,059 | CHRNB3 | 0.028 (0.0024), $7.7 \times 10^{-33}$ | 0.021 (0.0048), $1.6 \times 10^{-5}$ |
| rs56113850[b] (C) | 19:41,353,107 | CYP2A6 | 0.043 (0.0021), $4.0 \times 10^{-99}$ | 0.018 (0.0042), $2.1 \times 10^{-5}$ |
| rs1737894[b] (G) | 20:31,054,702 | NOL4L | 0.014 (0.0021), $9.9 \times 10^{-12}$ | 0.017 (0.0043), $1.1 \times 10^{-4}$ |

SNPs were associated with CPD at $P < 5 \times 10^{-8}$ in GSCAN ($N = 330,721$ biologically independent samples) and with ND at $P < 9.4 \times 10^{-4}$ ($\alpha = 0.05/53$ tests) in the cross-ancestry meta-analysis from iNDiGO ($N = 58,000$ biologically independent samples). Results are sorted by novelty (first three SNPs are previously unreported for ND) and then by iNDiGO p-values, and $\beta$ values correspond to direction of association for the effect alleles, with standard errors (SEs) shown.
NCBI National Center for Biotechnology Information.
[a]The locus flanking this SNP was not reported by prior GWAS of ND.
[b]SNP is located in a locus that was reported by prior GWAS of ND.

disorder[34] and its symptoms[31], posttraumatic stress disorder, and schizophrenia[35]) and smoking-related consequences (lung cancer and its histological subtypes[36] and coronary artery disease[37]). Among these positively correlated traits, $r_g$ values ranged from 0.18 (schizophrenia) to 0.75 (squamous cell lung cancer). Higher risk of ND was genetically correlated with lower age of smoking initiation[5] ($r_g = -0.55$) and fewer years of schooling[38] ($r_g = -0.34$).

For the traits with statistically significant genetic correlations with ND from the cigarette smoking, drug and alcohol use, personality, and psychiatric categories, we applied pairwise GWAS (GWAS-PW)[39] to identify shared genetic influences between FTND and each of these traits (Supplementary Fig. 5). GWAS-PW provides posterior probabilities for several models of genetic influence, including whether a given genomic region contains a variant that influences only ND (model 1), only the other trait (model 2), or both ND and the other trait (model 3). It also considers the scenario of whether the region contains a variant that influences ND and a separate variant influences the other trait (model 4). Both novel FTND-associated GWAS loci showed large probabilities for model 4 when comparing alcohol dependence and ND (posterior probabilities >0.97). The region surrounding rs2714700 also showed large model 4 probabilities for comparisons with depressive symptoms and schizophrenia. The region surrounding rs1862416 exhibited large model 3 probabilities for major depressive disorder and smoking initiation.

Rs1862416 was located within the boundaries of a genome-wide significant locus for smoking initiation (chr5:164,596,435-168,114,971), and to assess the independence of association signals at the single variant level, we performed conditional modeling using Genome-wide Complex Trait Analysis (GCTA)[40,41]. All 6 lead SNPs in this GSCAN-identified locus were in low LD with rs1862416 (maximum $r^2 = 0.0047$ (Supplementary Fig. 6), maximum D' = 0.46), and three were nominally associated with ND at $P < 0.05$ (Supplementary Table 4). Among our iNDiGO studies, rs1862416 remained associated with ND in models conditioned on each GSCAN lead SNP individually ($P = 7.9 \times 10^{-8}$–$1.8 \times 10^{-8}$) and with all 6 SNPs taken together ($P = 2.2 \times 10^{-7}$). Rs2714700 was located >1 MB away from any GSCAN-identified locus, so

conditional modeling was not necessary. Altogether, the GWAS-PW results suggest pleiotropy of smoking-related and comorbid traits in our two novel ND-associated regions, but at the variant level, the rs2714700 and rs1862416 associations with ND are independent of the GSCAN-identified variants.

**Regulatory annotations suggest target genes.** Credible set analysis of the chr7q21 locus narrowed the list of most likely causal variants to the lead SNP (rs2714700) and three others (rs2714674, rs1464692, and rs2707864) (Supplementary Table 8). Rs2714700, an intergenic SNP, is not a significant cis-eQTL with any gene-level expression in the Genotype-Tissue Expression (GTEx; v8) project, but it was implicated as a cis-eQTL for the MAGI2-AS3 transcript in hippocampus from BrainSeq[42] ($N = 551$; $P = 8.5 \times 10^{-4}$). The protective allele for ND (rs2714700-T) was associated with higher expression of the MAGI2-AS3 transcript ENST00000414797.5. Rs1464692 was also implicated as a cis-eQTL for the MAGI2-AS3 transcript in hippocampus from BrainSeq ($N = 551$; $P = 8.1 \times 10^{-4}$), and rs2707864 is located in a DNaseI hypersensitivity site in adult and fetal fibroblast cells in HaploReg[39] (Supplementary Table 8).

The lead SNP at the chr5q34 locus, rs1862416, is annotated to enhancer histone marks in brain (specifically, germinal matrix during fetal development and the developed prefrontal cortex, anterior caudate, and cingulate gyrus tissues) and several other tissues in HaploReg[43]. It is also located in the promoter of CTB-77H17.1, which is a novel antisense RNA transcript encoded within a TENM2 intron. In GTEx, rs1862416 was reported as a significant lung-specific cis-eQTL SNP for TENM2. The ND risk-conferring allele (rs1862416-T) was associated with decreased gene-level TENM2 expression in lung. CTB-77H17.1 was too lowly expressed across GTEx tissues to test its expression levels by rs1862416. Two additional, potentially causal variants identified in a credible set analysis were similarly annotated to enhancer and promoter markers in brain (prefrontal cortex, astrocyte) and fetal lung in HaploReg (rs36064369) and as lung-specific cis-eQTL in GTEx (rs116612101) (Supplementary Table 8).

**ND heritability is enriched for genes expressed in cerebellum**.
To assess whether heritability of ND is enriched in regions surrounding genes with the highest specific expression patterns in given tissue/cell type(s), we applied LDSC-SEG[44] using the EUR-specific ND GWAS meta-analysis results with reference to 205 tissues/cell types with publicly available gene expression data assembled from GTEx[45] (53 human tissues/cell types) and the underlying data that is used to comprise the Data-driven Expression Prioritized Integration for Complex Traits (DEPICT) tool[46,47] (152 tissues/cell types from humans and rodent models). We observed statistically significant enrichment in one tissue (cerebellum) at Bonferroni-corrected $P < 2.4 \times 10^{-4}$ (Supplementary Data 5), indicating that genes spanning ND-associated SNPs are enriched for specific expression in the cerebellum relative to other tissues/cell types.

**Combining evidence across FTND and HSI finds additional loci.** Given the strong genetic correlation between FTND and HSI ($r_g > 1$, Supplementary Table 7), we expanded our GWAS meta-analyses in the iNDiGO cohorts by additionally including the HSI-based UK Biobank results. The cross-ancestry GWAS meta-analysis ($N = 91,791$) is presented in Supplementary Figs. 7–8. Seven genome-wide significant loci were identified: *ARHGAP25* (chr2p13), *MAGI2/GNAI1* (chr7q21), *CHRNB3* (chr8p11), *FAM163B/DBH* (chr9q34), *AGPHD1* (nearby *CHRNA5*; chr15q25), *CYP2A6* (chr19q13), and *CHRNA4* (chr20q13) (see Supplementary Table 9 for lead SNP results). Of these, only the *ARHGAP25* locus, tagged by rs144481999, was previously unreported for any cigarette smoking phenotype. Although monomorphic among African Americans, rs144481999 is a low frequency (minor allele frequency [MAF] = 1.5%), but well-imputed (Rsq = 0.70–0.98), SNP in European ancestry cohorts, and the association of its T allele with increased risk is supported by several cohorts (e.g., $P < 0.05$ in deCODE, OZ-ALC, Yale-Penn, and UK Biobank).

## Discussion

We expanded current knowledge of ND in this largest GWAS to date, by identifying novel genome-wide significant loci as well as known loci, extending associations of additional loci implicated for other smoking phenotypes, and detecting significant genetic correlations of ND with 18 other complex phenotypes and with gene expression in cerebellum. The top novel SNPs between *MAGI2* and *GNAI1* (chr7q21), at *TENM2* (chr5q34), and at *ARHGAP25* (chr2p13) were independent of previously reported GWAS signals for any smoking trait. Three of our genome-wide significant loci were known: (1) *CHRNA5-CHRNA3-CHRNB4* (chr15q25) is irrefutably associated with ND, as driven largely by CPD[6]. (2) Our initial GWAS meta-analysis of 5 studies (now part of the iNDiGO consortium)[21] identified *CHRNA4* (chr20q13) at genome-wide significance. Subsequent associations were found with heavy vs. never smoking in the UK Biobank[48] and with initiation, CPD, and cessation in GSCAN[5]. (3) *DBH* (chr9q34) was first identified as genome-wide significant for smoking cessation but later associated with ND in our meta-analysis of 15 studies (now part of the iNDiGO consortium)[22] and with CPD and cessation in GSCAN[5]. Known loci were corroborated at the gene level with aggregated single SNP associations that take physical proximity and chromatin interactions or *cis*-eQTL evidence into account.

The novel ND-associated locus with lead SNP rs2714700 is intergenic between *MAGI2* (membrane associated guanylate kinase, WW and PDZ domain containing 2) and *GNAI1* (G protein subunit alpha i1). We identified rs2714700 at genome-wide significance for its association with ND, which was driven by CPD (unlike rs1862416), TTFC, and other FTND items, indicating that this SNP association may reflect both primary and secondary features of ND. Primary (or core) features of ND are necessary and sufficient for habit formation (heaviness of smoking [tolerance], automaticity, loss of control, and craving), while secondary features of ND underlie smoking that is goal based, e.g., relief of negative mood or cognitive enhancement[49–52]. Rs2714700 was also associated with HSI in the independent UK Biobank. The *cis*-eQTL evidence for rs2714700 in the hippocampus suggests that it may influence expression of the long noncoding RNA *MAGI2-AS3* (MAGI2 antisense RNA 3). *MAGI2-AS3* has been mainly studied for its role in the progression of cancer, including glioma in the brain[53]. No genome-wide significant associations have been reported within 1 MB of rs2714700 in the GWAS catalog. Our evidence of genome-wide significance for rs2714700 points to a novel locus that has not been associated with smoking or any related trait, and its functional relevance merits further investigation.

We also observed a genome-wide significant association of ND with rs1862416, a lung-specific *cis*-eQTL for *TENM2*. *TENM2* encodes teneurin transmembrane protein 2, a cell surface receptor that plays a fundamental role in neuronal connectivity and synaptogenesis[54]. With rs1862416 residing in the promoter of *CTB-77H17.1*, it could influence this antisense RNA, which in turn could dysregulate its sense transcript, *TENM2*. As an illustrative example, the SNP rs4307059, identified at genome-wide significance and independently replicated for autism[55], is annotated to and acts as a promoter region *cis*-eQTL for the antisense RNA *MSNP1AS* (moesin pseudogene 1, antisense) that influences regulation of its sense transcript, *MSN*[56]. However, while rs1862416 is generally indicated for its potential regulatory role (i.e., enhancer and promoter annotations and *cis*-eQTL evidence), its specific effect on either *CTB-77H17.1* or *TENM2* regulation in brain tissue was not evident in currently available data.

Further, independent association testing using HSI in the UK Biobank did not yield statistical significance for rs1862416. Similarly, the gene-based associations for the novel loci were not corroborated in UK Biobank. These differences in observed SNP- and gene-based associations may reflect components of ND that are not fully captured by the two FTND items that comprise the HSI (TTFC and CPD), as suggested by the specific FTND item association testing among the iNDiGO studies. Rs1862416 was suggestively associated ($P < 0.05$) with TTFC, "Which cigarette would you hate most to give up?" (the first one in the morning vs. all others), and "Do/did you smoke if you are so ill that you are in bed most of the day?" (yes/no). These item responses reflect withdrawal symptoms that are indicative of secondary features of ND, as compared with primary features of ND associated with habit formation[49–52]. Having the composite ND phenotype may have enhanced our power for discovering *TENM2*, but its detection in the UK Biobank may have been limited by the reliance on the HSI.

Beyond our discovery of rs1862416 with ND, SNPs across the *TENM2* gene have been identified at genome-wide significance, as presented in the GWAS catalog[57], for educational attainment[38], smoking initiation (ever vs. never smoking)[5,58–60], age of smoking initiation[5], smoking cessation (current vs. former smoking)[5], cigarette pack-years[61], alcohol consumption (drinks per week)[5], lung function[60,62], height[60], number of sexual partners[58], depression[63,64], risk taking tendency[58], body mass index[60], menarche (age at onset)[65], and regular attendance at a religious group[66]. Our pairwise comparisons supported pleiotropic associations in the *TENM2* region. At the variant level, all *TENM2* SNPs in the GWAS catalog have very low $r^2$ values with our novel SNP, rs1862416 (Supplementary Fig. 6), and our conditional modeling results showed that rs1862416 was associated with ND

independently from other *TENM2* SNPs implicated in GSCAN. While rs1862416 may have an ND-specific effect, the *TENM2* region has pleiotropic effects on ND, traits that are genetically correlated with ND, and other traits.

By combining ND GWAS results from iNDIGO with the highly correlated HSI results from the UK Biobank, our study also implicated the low frequency rs144481999, an intronic SNP in *ARHGAP25* (Rho GTPase Activating Protein 25). This finding merits independent replication testing and further biological characterization, as little is known about the regulatory potential of rs144481999 or its flanking *ARHGAP25* gene in relation to addiction.

The genetics of smoking behaviors, more broadly, has rapidly evolved with the GSCAN consortium having amassed a very large sample size and identified 298 genome-wide significant loci for smoking traits representing single components: ever vs. never smoking, age of smoking initiation, CPD, and current vs. former smoking[5]. We observed statistically significant genetic correlations of each of these smoking traits with ND (highest $r_g = 0.95$, as observed between ND and CPD), yet our two novel ND-associated loci were not identified at genome-wide significance by GSCAN (smallest $P = 0.033$ for rs1862416-T; smallest $P = 0.016$ for rs2714700-T), suggesting that these loci are specific to ND. Similarly, the majority of GSCAN-identified loci were trait-specific (191 of the 298 loci), whereas the other 107 loci were pleiotropic with associations identified for two or more of the smoking traits[5]. In our evaluation of GSCAN-identified loci, we corroborated associations of several previously implicated loci for ND (e.g., nicotine acetylcholine receptors genes *CHRNA5-A3-B4* and *CHRNA4*) and three additional loci (*DRD2*, *C16orf97*, and *CHRNB2*) that have not been reported in prior ND GWAS. Of these loci, *DRD2* is notable as a long-studied addiction candidate gene[4] and its recent identification as genome-wide significant for alcohol use disorder for rs4936277[67], which is correlated ($r^2 = 0.94$ in 1000 G EUR, 0.82 in 1000 G AFR) with rs7125588, the top SNP identified for CPD in GSCAN and associated with ND in iNDiGO; these results support a shared genetic effect of *DRD2* underlying addiction. Notably, rs7125588 is not correlated ($r^2 = 0.04$ in 1000 G EUR, 0.01 in 1000 G AFR) with the *DRD2* variant rs1800497 (historically referred to as the 'Taq1A' polymorphism), which is not significantly associated with ND in iNDiGO ($P = 0.24$).

Other GSCAN loci were detected for the single component smoking traits but show no evidence for association in our study (Supplementary Data 6), suggesting that these loci influence stages of smoking other than ND, or they exert weak effects on ND that we were underpowered to detect. We expect that additional GSCAN-identified loci are associated with ND, but their detection will require a larger sample size. These results demonstrate the utility of studying the genetics of the composite ND phenotype and comparing with GWAS of other smoking traits to tease apart loci that are specific to one stage (i.e., initiation, regular smoking, ND, cessation) vs. loci that influence multiple stages to better understand the full spectrum of smoking behaviors.

Beyond the smoking traits, we observed significant genetic correlations between ND and alcohol dependence[30], years of schooling[38], neuroticism[31], comorbid psychiatric traits (attention deficit hyperactivity disorder[32], bipolar disorder[33], major depression[34], schizophrenia[35], and posttraumatic stress disorder[68]) and smoking-related health consequences (lung cancer[36] and coronary artery disease[37]). Some of these observations corroborate prior findings (for example, alcohol dependence[30] and schizophrenia[69,70] with ND), whereas the other correlations extend to ND prior observations for the single component smoking traits (for example, CPD with years of schooling[5], neuroticism[5], major depression[5], coronary

artery disease[5], and lung cancer[36]). The genetic correlation between ND and gene expression in cerebellum is a notable observation consistent with cerebellum-specific *cis*-eQTL effects observed for the ND-associated *DNMT3B* SNP rs910083[22] and the age of smoking initiation-associated *CHRNA2* SNP rs11780471[36], both of which are also associated with lung cancer. These findings add to the evidence that the cerebellum may be important for ND risk[71,72], in addition to the other addiction-relevant brain tissues. However, since the cerebellum contains a higher neuronal concentration than other brain tissues[44,73], future studies are needed to decipher whether the cerebellar gene regulatory effects in the etiology of ND are due to neuronal activity. Additionally, although genetic correlation between ND and another trait suggest shared genetics underlying the phenotypes, multiple mechanisms can produce significant correlations (i.e., unmeasured intermediary phenotypes, correlated risk variants, mediation)[74–76]. Identifying the true mechanistic explanation requires further investigations.

In addition to the correlation between ND and cerebellar gene expression, the functional annotation of ND-associated loci using several gene-based analyses and enrichment tests implicated gene expression in multiple tissues and brain regions as relevant to ND (e.g., substantia nigra, hippocampus, and lung). Although it may appear that these methods produce discrepant results, it is important to consider the differences in reference data, model assumptions, and motivations underlying the methods for these analyses when interpreting findings. For example, S-MultiXcan provides the smallest p-values when using the single best tissue model (substantia nigra), but this does not indicate that the functional relevance of the ND-associated loci is exclusive to this brain region. The multi-tissue models also identified statistically significant ND associations with *PSMA4* and *CHRNA5*. Together these results highlight substantia nigra but also convey the relevance of other brain regions. Relatedly, the lack of substantia nigra evidence for BrainSeq eQTLs is not a discrepant finding. The reference data underlying the BrainSeq eQTLs only includes hippocampus and prefrontal cortex gene expression, thus this resource offers complementary information.

The present ND GWAS meta-analysis follows two prior waves of data assembly by the iNDiGO consortium ($Ns = 17,074$[21], $38,602$[22], and now $58,000$) and is the largest to date for the field. Despite still having substantially smaller sample sizes than the GSCAN GWAS, at each wave, increasing sample size for diverse ancestry groups (EURs and AAs) has illuminated ND-associated loci, some of which are shared with other stages of smoking while others are specific to ND. Our present findings underscore the complexity even within the ND phenotype, as our novel loci displayed patterns of association with specific FTND items that reflect primary or secondary ND features, e.g., the *TENM2* SNP influenced secondary features that are not captured simply by HSI. Future studies are needed to further dissect the genetic architectures underlying each of the specific FTND items. Understanding genetic similarities and differences that underlie these items and their contributions to primary vs. secondary ND may better inform treatment strategies, e.g., changing environmental cues for individuals whose smoking is driven solely by primary ND features vs. treating withdrawal for individuals whose ND is augmented with secondary features[51]. Studying the genetics of ND alongside other smoking traits (e.g., initiation and cessation) is key to gaining a better understanding of the neurobiological perturbations that influence the trajectory of smoking behaviors and their treatment implications.

## Methods
**Study overview**. We assembled 58,000 European ancestry or African American participants from 23 iNDiGO consortium studies with genome-wide SNP genotypes and FTND phenotype data available for ever smokers to perform ND GWAS

meta-analyses. Institutional review boards for the respective studies approved the study protocols, and all participants provided written informed consent. The meta-analysis combining the summary statistics from the 23 studies was approved by the RTI International Institutional Review Board. Fifteen of the studies were included from our prior GWAS using their original or updated sample sizes (total $N$ increased from 38,602[22] to 46,098 in the current analysis), while 8 studies were added for the current study (total $N = 11,902$). Participant characteristics are provided in Supplementary Table 2, and details of the study designs are provided in the Supplementary Methods. Ever smokers were defined by having reported smoking 100 or more cigarettes in their lifetime[21,22], unless otherwise stated in the study design description (see Supplementary Methods), and ND was defined by the FTND[12].

Our standard QC pipeline was applied to each study, unless otherwise stated in the study design description (see Supplementary Methods). Participants were removed due to genotype missing rate >3%, sample duplication (identity-by-state >90%), first-degree relatedness (identity-by-descent >40%), gender discordance ($F_{st} < 0.2$ for chromosome X single nucleotide polymorphisms (SNPs) to confirm females and $F_{st} > 0.8$ to confirm males), excessive homozygosity ($F_{st} > 0.5$ or $F_{st} < -0.2$), or chromosomal anomalies. SNPs were removed due to missing rate >3% across samples or Hardy–Weinberg equilibrium (HWE) $P < 1 \times 10^{-4}$. Genotyped SNPs passing QC were used as input for imputation with reference to 1000 G phase 3 across all studies[4], except for deCODE which used an Icelandic whole genome sequencing-based reference panel[77].

**FTND and categorical ND definitions for discovery GWAS.** The FTND[12] is a well-validated, widely used questionnaire that assesses psychologic dependence on nicotine using the following six items:

(1) How soon after you wake up do/did you smoke your first cigarette? Categorical responses: within 5 min, 6–30 min, 31–60 min, or after 60 min.
(2) Do/Did you find it difficult to refrain from smoking in places where it is forbidden, e.g., in church, at the library, in a cinema, etc.? Binary response: yes or no.
(3) Which cigarette would you hate most to give up? Binary response: the first one in the morning or all others.
(4) How many cigarettes per day do/did you smoke? Categorical response: 10 or less, 11–20, 21–30, or 31 or more.
(5) Do/did you smoke more frequently during the first hours after waking than during the rest of the day? Binary response: yes or no.
(6) Do/did you smoke if you are so ill that you are in bed most of the day? Binary response: yes or no.

Additional details on the protocol and scoring algorithm are provided in the PhenX Toolkit[78], a catalog of commonly ascertained phenotype and exposure measures: https://www.phenxtoolkit.org/protocols/view/31001. Briefly, FTND scores range from 0 (no dependence) to 10 (highest dependence level)[23,79]. FTND can be administered on current or former smokers based on the time period when they reported smoking the most (i.e., lifetime FTND) or among current smokers based around the time of interview (i.e., current FTND). We used lifetime FTND collected among current and former smokers in AAND, ADAA, EAGLE, COGEND, COGEND2, COPDGene2, deCODE, eMERGE, FINRISK, FTC, GAIN, JHS/ARIC, MCTFR, nonGAIN, NTR, OZ-ALC, SAGE*, Spit for Science, and Yale-Penn. We used current FTND that was available in COHRA1, COPDGene, eMERGE, German, and UW-TTURC. We previously found only small differences in genetic association results due to any measurement variance when using current vs. lifetime FTND[80].

We used the FTND to derive a categorical variable for ND[21,22]: scores of 0–3 for mild, 4–6 for moderate, and 7–10 for severe. We relied solely on the FTND to define ND, except in two of the 23 studies (deCODE and JHS/ARIC) where we included smokers with FTND data available as well as low-intensity smokers who had only CPD data that we used as a proxy to define mild dependence (CPD ≤ 10)[21,22]. Our prior assessment showed high concordance of CPD ≤ 10 and FTND scores 0–3 (86.4%)[22], suggesting that CPD can be used to define mild dependence with little phenotype misclassification. However, any phenotype misclassification would be expected to conservatively bias results, leading to reduced statistical power, attenuated effect size estimates, and thus underestimate SNP associations with ND[81,82]. Moderate and severe dependence was defined solely by FTND scores across all studies, as our prior assessment showed lower concordance between FTND and CPD for defining these categories[22].

**ND GWAS meta-analysis.** For each study, genome-wide SNP/indel associations with the 3-level categorical ND outcome were tested within an ancestry group using linear regression. Covariates included age, sex, principal component eigenvectors, and study-specific covariates where warranted. For studies that included relatives, relatedness was accounted for in the regression modeling. See Supplementary Methods for additional study-specific details.

GWAS results were combined using fixed-effect inverse variance-weighted meta-analyses in METAL[83]. Prior to performing meta-analyses, we applied genomic control to results from one study, deCODE, to adjust for inflation due to relatedness among participants ($\lambda = 1.12$); all other studies had low inflation ($\lambda = 0.99$–$1.04$) (Supplementary Table 2). We removed SNPs/indels with MAF <

1% in the 1000 G phase 3 reference panel for the analyzed ancestry group (1000 G European or African superpopulations), imputation info score < 0.3, or availability in only one study. All variant annotations correspond to the National Center for Biotechnology Information (NCBI) build 37. The threshold of genome-wide significance was set at $P = 5 \times 10^{-8}$ [22]. Regional association plots of novel genome-wide significant loci were constructed using LocusZoom[84] with references of either 1000 G European or African panels to estimate linkage disequilibrium of the lead SNP (based on smallest meta-analysis P-value) and surrounding SNPs. The lead SNP for each novel FTND locus was tested for association with each of the specific FTND items.

For any ND-associated SNPs located within the bounds of loci identified by GSCAN (1 MB surrounding the lead SNP)[5], conditional models were analyzed using our GWAS summary statistics and the Genome-wide Complex Trait Analysis (GCTA) tool, adjusted for the lead SNPs in GSCAN[40,41]. To contextualize the magnitude of the observed effect sizes, we calculated odds ratios (ORs) using the $\beta$ estimate from the single SNP linear regression model (OR = $\exp[2 \times \beta_{SNP}]$ for severe vs. mild ND, with OR > 1 corresponding to an increased risk of severe ND) and compared these values across studies and ancestries using the Forest Plot Viewer[85].

In follow-up analyses of lead SNPs from novel loci, we tested associations of each specific FTND item, using linear regression models for items with categorical responses (items #1 and 4) and logistic regression models for items with binary responses (items #2, 3, 5, and 6), followed by meta-analysis of results across studies. Due to varying genotype and phenotype data availability for the novel loci and specific FTND items, some studies could not utilize the full sample set for specific FTND item testing. These studies include AAND, COGEND, COGEND2, deCODE, Dental Caries, GAIN, German, JHS/ARIC, and nonGAIN[5].

**Independent testing using heaviness of smoking index.** Novel, genome-wide significant SNPs from our ND GWAS meta-analysis were tested in the UK Biobank. Since there are no other ND datasets with comparably large sample sizes, we relied on the HSI that is available in the UK Biobank. The UK Biobank collected data on two of the 6 FTND items (CPD and time to first cigarette in the morning [TTFC]) among current smokers, who reported smoking on most or all days. These two items together constitute the HSI, which has historically been considered a suitable proxy for the full-scale FTND[12]. To evaluate the agreement in our FTND categories (score range = 0–10; mild [scores 0–3], moderate [scores 4–6], and severe [scores 7–10] as we have routinely used before[21,22]) with HSI categories (score range = 0–6; mild [scores 0–2], moderate [scores 3–4], and severe [scores 5–6], in accordance with the scoring algorithm by the American Society of Clinical Oncology[86]), we used EURs in COGEND, which was ascertained specifically for ND. We compared FTND and HSI categories based on lifetime FTND (i.e., FTND based on time smoked most). Results are presented in Supplementary Table 10. Concordance was highest for mild dependence at 94.9%; i.e., among those defined as having mild dependence by the HSI, 94.87% also had mild dependence as defined by the full-scale FTND. We observed concordance of 81.2% for moderate dependence and 84.9% for severe dependence. Overall concordance was high (89.3%), corroborating the utility of the HSI categories as a proxy for defining ND in the UK Biobank. Genome-wide significant SNP associations from our GWAS meta-analysis were tested for association using 33,791 current smokers with HSI data available in UK Biobank: 18,063 mild (HSI scores 0–2), 13,395 moderate (HSI scores 3–4), and 2,333 severe (scores 5–6) dependence. This final analysis data set included only unrelated individuals, as we removed 844 third-degree or closer relatives prior to analysis; for each related pair/cluster, individuals who had more relatives and who were light smokers were prioritized for removal. For our SNP-HSI association testing, we followed the model employed by systemic GWAS analyses for a multitude of phenotypes (see http://www.nealelab.is/uk-biobank/), adjusting for the following covariates: sex, age, age$^2$, age × sex, age$^2$ × sex, and PC eigenvectors, with the age$^2$ and interaction terms among the age and sex variables intended to account for non-linear associations. To compare the genetic correlation of ND with HSI, we restricted UK Biobank analyses to the 31,854 current smokers of self-reported European ancestry.

**Gene-based association testing.** To assess evidence for association beyond single variants, we applied two methods that aggregate SNP-based summary statistics at the gene level. For genome-wide testing with both methods, we used the EUR-specific GWAS meta-analysis results from iNDiGO as the input dataset, given the reliance on linkage disequilibrium (LD) reference data by ancestry in calculating the gene-based summary statistics. First, H-MAGMA[27] computes gene-based association statistics by aggregating SNP associations based on physical proximity to the target gene(s) measured by chromatin interaction maps from human brain tissue. We included SNPs with an rs identification number (9,525,836 SNPs) and coupled them with Hi-C reference datasets from fetal[87] and adult brain tissues, specifically cortical tissues[88], that are available for running H-MAGMA. H-MAGMA converted SNP-level p-values into gene-level p-values. We identified statistically significant genes that were associated with ND at a Bonferroni-corrected threshold of $P < 2.7 \times 10^{-6}$ ($\alpha = 0.05/18,655$ protein coding genes).

Second, we applied Summary-MultiXcan (S-MultiXcan)[28] to compute gene-level associations by leveraging imputed, genetically driven gene expression using RNA-Seq across the 13 adult brain tissues in GTEx as reference data. S-

MultiXcan[28], an extension of the S-PrediXcan method for integrating eQTLs with GWAS summary statistics[89], aggregates eQTL information across multiple tissue types to enhance statistical power, while still presenting the single tissue with the best evidence for association. We applied Bonferroni correction to declare statistically significant gene-based associations as $P < 3.5 \times 10^{-6}$ ($\alpha = 0.05/14,494$ genes).

For both gene-based methods, we carried forward significant gene-level associations and tested them in the UK Biobank, using HSI as a proxy for ND, as done with the single SNP associations.

**Cross-trait genetic correlations with ND**. Summary statistics from the EUR-specific meta-analyses were used as input into LD score regression (LDSC)[29] with reference to the 1000 G EUR panel to estimate the SNP heritability ($h_g^2$) of ND and its genetic correlations with 47 other complex phenotypes, including other smoking, drug, and alcohol use and dependence traits, smoking-related health consequences (e.g., cancer, COPD, and coronary heart disease), psychiatric and neurologic disorders, cognitive and educational traits, and brain volume metrics. The full list of phenotypes and GWAS datasets, as obtained from LD Hub[90] or shared by the original study investigators, are provided in Supplementary Table 7.

**Tissue and cell type gene expression enrichment of ND loci**. EUR-specific GWAS meta-analysis summary statistics were input into stratified LDSC, as applied to specifically expressed genes (LDSC-SEG)[44], with reference to 205 tissues and cell types from two sources—RNA-sequencing data on 53 human tissues/cell types in GTEx[91] and array-based data on 152 tissues/cell types from humans and rodent models that underlie the DEPICT tool and made available in Gene Expression Omnibus[46,47]. See full list of the 205 tissues/cell types in Supplementary Data 5. Similarly to the initial application of LDSC-SEG[44], these two sources were selected because their expression data included a wide range of ND-relevant and other tissues and cell types in humans, as opposed to focused information on a particular tissue. LDSC-SEG involved comparing expression of each gene in each tissue/cell type with that in other tissues/cell types, selecting the top 10% of differentially expressed genes, annotating SNPs from the GWAS summary statistics that lie within 100 kilobase windows of the selected genes, and using the stratified LDSC method to estimate the enrichment in SNP heritability for ND for the given gene set compared to the baseline LDSC model with all genes. For each analysis, a Bonferroni correction was applied to assess statistical significance: $P < 0.0011$ ($\alpha = 0.05/47$ phenotypes) for LDSC and $P < 2.4 \times 10^{-4}$ ($\alpha = 0.05/205$ tissues/cell types) for LDSC-SEG.

**Associations of ND loci with other complex traits**. We applied pairwise GWAS (GWAS-PW v0.21 [github.com/joepickrell/gwas-pw/])[39] to characterize the cross-phenotype associations for ND and its genetically correlated phenotypes, as revealed in the LDSC analyses. Specifically, we applied GWAS-PW to the "Cigarette smoking", "Drug and alcohol use", "Personality", and "Psychiatric" phenotypes with significant genetic correlation with ND. Using EUR-specific GWAS summary statistics for ND and its correlated phenotypes, for each pairwise comparison of ND to a given phenotype, we calculated a correlation statistic used by GWAS-PW to account for potential sample overlaps between studies. Specifically, we applied fgwas v0.3.6 (https://github.com/joepickrell/fgwas), with genome-wide predefined LD blocks (https://bitbucket.org/nygcresearch/ldetect-data/src/master/EUR/fourier_ls-all.bed), to variants that have summary statistics for both phenotypes. Z-scores for all variants within genomic segments with a poster probability less than 0.2 for either phenotype were used to calculate a Pearson correlation coefficient. We then further reduced the SNP set to only SNPs with summary statistics available from both studies and that also were located within an LD block for any of the 5 FTND GWAS significant loci. For this step, we defined the LD blocks by using LDproxy[92] with the top (i.e., most significant) SNP from each FTND-associated locus and extracting $r^2$ values (based on 1000 Genomes Phase 3 EUR populations) for all SNPs within 0.5 MB of the top SNP. The minimum and maximum genomic coordinate for all extracted SNPs with $r^2 > 0.2$ were used as the LD block boundaries.

**cis-eQTL assessment of novel ND-associated SNPs**. For each novel locus, we identified a credible set, or the set of SNPs most likely to contain the causal variant, using a Bayesian method[93] implemented via LocusZoom[84]. To assess evidence for SNP-gene associations, SNPs in the credible set were queried against GTEx (version 8) cis-eQTL results derived from SNP genotype and RNA-sequencing data across 44 tissues ($N = 126–209$ for the 13 brain tissues)[91]. The GTEx portal (https://gtexportal.org/home/) presents significant single-tissue cis-eQTLs, based on a false discovery rate (FDR) < 5%.

We also assessed single-tissue cis-eQTL evidence from the BrainSeq consortium that includes larger sample sizes with SNP genotype and RNA-sequencing data available in two brain tissues, dorsolateral prefrontal cortex ($N = 453$) and hippocampus ($N = 447$)[42]. Of the 551 individuals with data available in at least one brain tissue, 286 were schizophrenia cases; case/control status was included as a covariate for adjustment in the cis-eQTL analysis[94] Significant cis-eQTLs at FDR < 10% are available at http://eqtl.brainseq.org/phase2/eqtl/.

**Reporting summary**. Further information on research design is available in the Nature Research Reporting Summary linked to this article.

## Data availability

The prior meta-analysis summary statistics[22] are available via dbGaP accession number phs001532.v1.p1. The summary statistics generated from the current study are included under version 2 of this dbGaP study with accession number phs001532.v2.p1. These summary statistics are also available upon reasonable request to the corresponding author (D.B.H.). Individual-level genotype and phenotype data for many of the contributing studies are available via dbGaP, as outlined in the study descriptions in the Supplementary Methods. The dbGaP accession numbers for these studies are phs000092.v1.p1, phs000404.v1.p1, phs000095.v2.p1, phs000765.v1.p2, phs000093.v2.p2, phs000170.v2.p1, phs000021.v3.p2, phs000167.v1.p1, phs000286.v3.p1, phs000090.v1.p1, phs000277.v1.p1, phs000092.v1.p1, and phs000404.v1.p1. 1000 Genomes Phase 3 reference panel data are available at ftp://ftp.1000genomes.ebi.ac.uk/vol1/ftp/release/20130502/. GSCAN summary statistics are available at https://genome.psych.umn.edu/index.php/GSCAN. GWAS summary statistics from LD Hub are available at http://ldsc.broadinstitute.org/gwasshare/. LDSC EUR LD scores are available at ftp://atguftp.mgh.harvard.edu/brendan/1k_eur_r2_hm3snps_se_weights.RDS. LD blocks used with GWAS-PW are available at https://bitbucket.org/nygcresearch/ldetect-data/src/master/EUR/fourier_ls-all.bed. BrainSeq Consortium cis-eQTL summary statistics are available at http://eqtl.brainseq.org/phase2/eqtl/.

## Code availability

All software used to perform these analyses is available online.

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

## Acknowledgements

This work was supported by the National Institute on Drug Abuse grant numbers R01 DA042090 (PI: DBH) and R01 DA036583 (PI: LJB) and by National Cancer Institute grant number U19 CA203654 (LJB; PI: Amos). The authors thank deCODE Genetics / Amgen and its investigators (Gunnar W. Reginsson, Thorgeir E. Thorgeirsson, and Kari Stefansson) for their data contributions, which were supported in part by NIDA R01 DA017932 (PI: Kari Stefansson). Acknowledgments for all other ND studies, which were contributed by the authors and/or made publicly available, are included in Supplementary Note 1. UK Biobank Resource data were obtained under Application Number 24603.

## Author contributions

Contributions of each author are categorized using terms from the Contributor Roles Taxonomy (https://casrai.org/credit/). Conceptualization: D.B.H., D.W.M., E.O.J., L.J.B., M.J.B., M.R., N.E.C., T.B.B.; Data curation: B.C.Q., C.C.M., F.A., J.H., J.K., M.N., N.C.G., N.E.C., N.L.S., R.G., S.Z., Y.G.; Formal analysis: A.W., B.C.Q., C.A.M., C.C.M., D.D., F.A., H.Y., J.A.M., J.H., M.J.B., M.N., M.U.C., N.C.G., R.S., T.P., Y.G.; Funding acquisition: D.B.H., D.M.D., D.W.M., E.O.J., J.K., L.J.B., M.R., W.G.I.; Investigation: D.I.B., E.O.J., F.A., G.W., H.R.K., J.G., J.H., J.K., J.M.V., L.A.F., L.J.B., M.C.N., M.J.B., M.M., M.R., M.T.L., N.L.S., P.M., S.V., W.G.I.; Methodology: D.W.M.; Project administration: D.B.H., D.M.D., D.W.M., J.K.; Resources: D.B.H., D.I.B., D.M.D., D.W.M., G.W., J.G., J.K., J.M.V., K.A.Y., L.A.F., L.J.B., M.C.N., M.J.B., M.L.M., M.M., M.R., M.T.L., N.C.G., N.E.C., P.M., R.G., S.V., W.G.I.; Software: A.W., B.C.Q., H.W., J.A.M., M.U.C., N.C.G., N.Y.A.S.; Supervision: D.B.H., D.I.B., D.M.D., D.W.M., E.O.J., J.K., M.J.B., M.N.; Validation: D.B.H., M.L.; Visualization: B.C.Q., D.B.H., J.A.M., M.J.B.; Writing – original draft: B.C.Q., D.B.H., J.A.M., S.Z., T.B.B.; Writing—review & editing: A.W., B.C.Q., C.A.M., C.C.M., D.B.H., D.I.B., D.M.D., D.W.M., E.O.J., F.A., G.W., H.R.K., J.E.H., J.G., J.K., J.M.V., K.A.Y., L.A.F., L.J.B., M.C.N., M.J.B., M.L.M., M.M., M.R., M.T.L., N.C.G., N.E.C., N.L.S., P.M., S.V., T.B.B.

## Competing interests

L.J.B. and the spouse of N.L.S. are listed as inventors on U.S. Patent 8,080,371, 'Markers for Addiction' covering the use of certain SNPs in determining the diagnosis, prognosis, and treatment of addiction. Y.G. is an employee of GeneCentric Therapeutics. Although unrelated to this research, H.R.K. is an advisory board member for Dicerna and a member of the American Society of Clinical Psychopharmacology's Alcohol Clinical Trials Initiative, which was supported in the last 3 years by AbbVie, Alkermes, Ethypharm, Indivior, Lilly, Lundbeck, Otsuka, Pfizer, Arbor and Amygdala Neurosciences. H.R.K. and J.G. are named as inventors on PCT patent application #15/878,640 entitled: "Genotype-guided dosing of opioid agonists," filed January 24, 2018. J.K. consulted for Pfizer in 2012–2015 on ND. The remaining authors declare no competing interests.

## Additional information

Bryan C. Quach[1], Michael J. Bray[2], Nathan C. Gaddis[1], Mengzhen Liu[3], Teemu Palviainen[4], Camelia C. Minica[5], Stephanie Zellers[3], Richard Sherva[6], Fazil Aliev[7,8], Michael Nothnagel[9,10], Kendra A. Young[11], Jesse A. Marks[1], Hannah Young[3], Megan U. Carnes[1], Yuelong Guo[1,12], Alex Waldrop[1], Nancy Y. A. Sey[13], Maria T. Landi[14], Daniel W. McNeil[15,16], Dmitriy Drichel[9,10], Lindsay A. Farrer[6,17,18,19,20], Christina A. Markunas[1], Jacqueline M. Vink[21], Jouke-Jan Hottenga[5], William G. Iacono[3], Henry R. Kranzler[22,23], Nancy L. Saccone[24,25], Michael C. Neale[26,27], Pamela Madden[2], Marcella Rietschel[28], Mary L. Marazita[29], Matthew McGue[3], Hyejung Won[13], Georg Winterer[30], Richard Grucza[31], Danielle M. Dick[7,32,33], Joel Gelernter[34,35,36,37], Neil E. Caporaso[38], Timothy B. Baker[39], Dorret I. Boomsma[5], Jaakko Kaprio[4,40], John E. Hokanson[11], Scott Vrieze[3], Laura J. Bierut[2], Eric O. Johnson[1,41] & Dana B. Hancock[1]✉

[1]GenOmics, Bioinformatics, and Translational Research Center, Biostatistics and Epidemiology Division, RTI International, Research Triangle Park, NC 27709, USA. [2]Department of Psychiatry, Washington University, St. Louis, MO 63130, USA. [3]Department of Psychology, University of Minnesota Twin Cities, Minneapolis, MN 55455, USA. [4]Institute for Molecular Medicine Finland (FIMM), University of Helsinki, 00290 Helsinki, Finland. [5]Department of Biological Psychology, Vrije Universiteit, 1081 BT Amsterdam, The Netherlands. [6]Department of Medicine (Biomedical Genetics), Boston University School of Medicine, Boston, MA 02118, USA. [7]Department of Psychology, Virginia Commonwealth University, Richmond, VA 23284, USA. [8]Faculty of Business, Karabuk University, 78050 Kılavuzlar/Karabük Merkez/Karabük, Turkey. [9]Cologne Center for Genomics, University of Cologne, 50931 Köln, Germany. [10]University Hospital Cologne, 50931 Köln, Germany. [11]Department of Epidemiology, University of Colorado Anschutz Medical Campus, Aurora, CO 80045, USA. [12]GeneCentric Therapeutics, Research Triangle Park, NC 27709, USA. [13]Department of Genetics, University of North Carolina at Chapel Hill, Chapel Hill, NC 27514, USA. [14]Genetic Epidemiology Branch, Division of Cancer Epidemiology and Genetics, National Cancer Institute, National Institutes of Health, United States Department of Health and Human Services, Bethesda, MD 20892, USA. [15]Department of Psychology, West Virginia University, Morgantown, WV 26505, USA. [16]Department of Dental Practice and Rural Health, West Virginia University, Morgantown, WV 26505, USA. [17]Department of Neurology, Boston University School of Medicine, Boston, MA 02118, USA. [18]Department of Ophthalmology, Boston University School of Medicine, Boston, MA 02118, USA. [19]Department of Epidemiology, Boston University School of Public Health, Boston, MA 02118, USA. [20]Department of Biostatistics, Boston University School of Public Health, Boston, MA 02118, USA. [21]Behavioural Science Institute, Radboud University, 6500 HE Nijmegen, The Netherlands. [22]Department of Psychiatry, University of Pennsylvania Perelman School of Medicine, Philadelphia, PA 19104, USA. [23]VISN 4 MIRECC, Crescenz VA Medical Center, Philadelphia, PA 19104, USA. [24]Department of Genetics, Washington University, St. Louis, MO 63130, USA. [25]Division of Biostatistics, Washington University, St. Louis, MO 63130, USA. [26]Virginia Institute for Psychiatric and Behavioral Genetics, Virginia Commonwealth University, Richmond, VA 23284, USA. [27]Department of Psychiatry, Virginia Commonwealth University, Richmond, VA 23284, USA. [28]Department of Genetic Epidemiology in Psychiatry, Central Institute of Mental Health, Medical Faculty Mannheim, University of Heidelberg, 68159 Mannheim, Germany. [29]Center for Craniofacial and Dental Genetics, Department of Oral Biology, University of Pittsburgh, Pittsburgh, PA 15261, USA. [30]Experimental & Clinical Research Center, Department of Anesthesiology and Operative Intensive Care Medicine, Charité - University Medicine Berlin, 10117 Berlin, Germany. [31]Departments of Family and Community Medicine and Health and Clinical Outcomes Research, Saint Louis University, St. Louis, MO 63130, USA. [32]College Behavioral and Emotional Health Institute, Virginia Commonwealth University, Richmond, VA 23284, USA. [33]Department of Human & Molecular Genetics, Virginia Commonwealth University, Richmond, VA 23284, USA. [34]Department of Psychiatry, Yale University School of Medicine, New Haven, CT 06511, USA. [35]Department of Genetics, Yale University School of Medicine, New Haven, CT 06511, USA. [36]Department of Neuroscience, Yale University School of Medicine, New Haven, CT 06511, USA. [37]Department of Psychiatry, VA CT Healthcare Center, West Haven, CT 06511, USA. [38]Occupational and Environmental Epidemiology Branch, Division of Cancer Epidemiology and Genetics, National Cancer Institute, National Institutes of Health, United States Department of Health and Human Services, Bethesda, MD 20892, USA. [39]Center for Tobacco Research and Intervention, Department of Medicine, University of Wisconsin School of Medicine and Public Health, Madison, WI 53726, USA. [40]Department of Public Health, Faculty of Medicine, University of Helsinki, 00290 Helsinki, Finland. [41]Fellow Program, RTI International, Research Triangle Park, NC 27709, USA. ✉email: dhancock@rti.org

