## [Peer Review File · Nature Communications]

Reviewers' comments:

Reviewer #1 (Remarks to the Author):

This is a nicely done study on the genetics of tobacco dependence. The authors give a clear presentation of a GWAS conducted in the largest sample currently available. The manuscript is well structured and well written. The main question I have after reading this paper, is whether conducting genetic analyses of nicotine dependence provides sufficiently unique information beyond a genetic study of other smoking phenotypes, such as the number of cigarettes smoked per day. Provided that the genetic correlation between nicotine dependence and cigarettes smoked per day is very high, with a point estimate of .95 (is it significantly different from 1?), I wonder whether we should focus on CPD only, since it is relatively easy to assess and much larger sample sizes are available than for nicotine dependence. Indeed, the number of significant loci in the current study is relatively small. On the other hand, we would not have a reliable estimate of the genetic correlation between nicotine dependence and CPD if this study would not have been conducted. Furthermore, the authors claim that they found significant associations in genes that were not previously found to be associated with CPD. Therefore, my overall conclusion is that this is a sound study with findings that are important for the field.

Some suggestions for improvement:

- Figure shows the genetic correlations with external traits but it is difficult to see whether CIs overlap 1. I think this would be interesting information to add, e.g., by adding a vertical line at $r_g=1$. The authors could also formally test which genetic correlations are not significantly lower than 1 after correction for multiple testing
- Did the authors consider to add PrediXcan analyses to test whether imputation of gene expression identifies additional loci beyond single SNP eQTL analysis? I was actually quite surprised that they did not conduct any type of gene-based testing (e.g., MAGMA)
- Page 9, paragraph on gene expression. Can the authors provide more information on which resources have been used?
- The authors could do more to describe the genetic overlap in terms of pleiotropy, e.g., pairwise GWAS (Pickrell et al) or MiXeR (Frei et al) to identify loci that are unique for a specific trait vs. loci that are shared across traits. Genomic SEM would be very useful to explore the genetic factor structure of the FTND items. I think additional analyses will be important since the number of novel loci is relatively limited.

Reviewer #2 (Remarks to the Author):

Review of the manuscript "Expanding the genetic architecture of nicotine dependence and its shared genetics with multiple traits: findings from the nicotine dependence GenOmics (INDiGO) Consortium", by Quach et al.

The authors report results from a GWAS meta-analysis across European and African ancestries of nicotine dependence and identify five genome-wide significant loci of which two are novel. The study provides new information to the field and deserves publication. However I have some concerns regarding the functional annotation and at some places important information is missing or is wrong, especially in the genetic correlation analyses.

My points that need to be addressed:

Functional annotation, page 8-9, the authors concentrate their functional annotation to the two index variants in the novel genome-wide significant loci. However, the causal variant is not necessary the index variant of a locus but could be any variant in high LD with the index variant. The authors should perform functional annotation of all variants/or a set of credible variants in the two regions and preferably use a fine mapping method that takes ancestry into account (e.g the method by Lam et al. Nature Genetics 2019)

The authors should provide more information about the phenotype tested already in the beginning

of the result section. The authors write in the introduction that FTND provides information about a composite phenotype. It should therefore be clear already from the start of the result section (not only the methods section) how the authors have defined the phenotype used in the GWAS meta-analysis.

Additionally, it is confusing how they have defined ND based on the FTND scores. In the result section page 5, bottom, the authors write that they have tested severe vs. mild ND, but in the method section page 16, they write that they have tested for association with a 3-level categorical ND. Please clarify this issue.

The authors test for genetic correlation with 45 complex phenotypes. For some of the phenotypes analyzed the authors refer to papers with outdated data in Supplementary Table 3. It is therefore not clear if the data they have used should be updated or the references are wrong. I have not gone through all references but can see that the references to ADHD is outdated (should instead be Demontis et al. 2019), Autism spectrum disorders (should instead be Grove et al. 2019), and bipolar disorder is outdated as well. Please clarify these analyses.

Minor corrections:

Page 5: "AA-specific" has not previously been defined.

Page 6: Please clarify if there are any sample overlap between GSCAN and iNDiGO samples

Page 6 bottom: I would prefer if you replace "For independent testing" with "For replication in an independent sample"

Page 6, bottom: Please already here describe the difference between HSI and FTND. For someone not familiar with these measures it would be nice know at this point that HSI comprise TTFC and CPD.

Page 7, top: for consistency replace "all discovery samples" with "iNDiGO samples"

Page 7, second paragraph: "...the factors of ND". This wording is unspecific, please specify what is meant by factors.

Page 7, second paragraph: "TTFC" is mentioned for the first time. Specify what it means.

Page 9, second paragraph: "To assess the enrichment of the EUR-specific ND..." I assume it is enrichment in the heritability, but this is not clear.

Page 10, second paragraph: The authors mention "primary and secondary features of ND". This is defined later in the discussion, but it would help to have this information already at this point.

Page 11 top: rs4307059 is not genome-wide significantly associated with autism.

Figure 2, item 3: "Cigarette most hated to give up", this description does not provide sufficient information about what the variant is associated with.

Page 1, second paragraph: "...yet despite the nearly complete sharing between ND and CPD.." this sentence is not clear. By "nearly complete sharing" I assume the authors mean a genetic correlation close to one, but this is not clear, please rephrase.

Page 12, second paragraph: "These observations resemble previously reported patters of genetic correlation between alcohol dependence and alcohol consumption". I do not agree with this statement. First of all, alcohol consumption and alcohol dependence disorder demonstrate opposite genetic correlations with several traits including educational attainment where the former is positively correlated and the latter negatively. Both CPD and ND demonstrate negative genetic correlation with educational attainment. Additionally, just because some loci are ND specific compared to CPD, it does not support the existence of similar pattern as the one observed at the polygenic level for alcohol dependence vs alcohol use disorder.

Page 13, first paragraph: what is Taq1A – if it is a gene, it should be in italics.

Page 13, third paragraph: insert references for the disorders included in the genetic correlation analyses.

Page 16, third paragraph: Please provide information about filtering on imputation info score, and have variants in the GWAS meta-analysis been filtered based on the effective sample size?

We appreciate the reviewers' thoughtful comments. We believe that addressing their comments (shown in italics), as detailed point-by-point below, has strengthened our study's results and conclusions. In addition, since the time of the original review, we have made our genome-wide meta-analysis summary statistics freely available to the scientific community via our dbGaP parent study phs001532: version 1 corresponding to our prior GWAS meta-analysis¹ and version 2 corresponding to the current meta-analysis (preview site available at https://www.ncbi.nlm.nih.gov/projects/gapprev/gap/cgi-bin/preview1.cgi?GAP_phs_code=xDGfFkzCFNLwpojP). These results sharing options are outlined in the "Data Availability" section of the main text (lines 493–496).

Reviewer #1's Comments

1. This is a nicely done study on the genetics of tobacco dependence. The authors give a clear presentation of a GWAS conducted in the largest sample currently available. The manuscript is well structured and well written. The main question I have after reading this paper, is whether conducting genetic analyses of nicotine dependence provides sufficiently unique information beyond a genetic study of other smoking phenotypes, such as the number of cigarettes smoked per day. Provided that the genetic correlation between nicotine dependence and cigarettes smoked per day is very high, with a point estimate of .95 (is it significantly different from 1?), I wonder whether we should focus on CPD only, since it is relatively easy to assess and much larger sample sizes are available than for nicotine dependence. Indeed, the number of significant loci in the current study is relatively small. On the other hand, we would not have a reliable estimate of the genetic correlation between nicotine dependence and CPD if this study would not have been conducted. Furthermore, the authors claim that they found significant associations in genes that were not previously found to be associated with CPD. Therefore, my overall conclusion is that this is a sound study with findings that are important for the field.

Response: We thank the reviewer for the compliments on the conduct and presentation of our study. The reviewer is correct that the FTND and other dependence measures are meaningfully correlated with cigarettes per day (CPD), so this is a very apt question. We have inserted the following material to address the reviewer's concern (lines 35–44).

While CPD is associated with key markers of ND, such as cessation likelihood², the FTND conveys additional valuable information. It is meaningfully associated with tobacco use diagnostic criteria from the *Diagnostic and Statistical Manual of Mental Disorders*³ and is more highly associated with withdrawal severity than is CPD.⁴ Its validity may be due to the inclusion of the time-to-first-cigarette in the morning (TTFC) item, which appears to be especially strongly associated with relapse likelihood⁵⁻⁷ and may be an especially informative measure of heritability of ND.⁸ Thus, the FTND provides somewhat different information than CPD and has been relatively understudied from a genetic perspective because of its more limited availability across datasets.

2. Figure shows the genetic correlations with external traits but it is difficult to see whether CIs overlap 1. I think this would be interesting information to add, e.g., by adding a vertical line at $r_g=1$. The authors could also formally test which genetic correlations are not significantly lower than 1 after correction for multiple testing.

Response: In revising Figure 3, to address comment #10 below, we added a vertical line at $r_g=1$ as requested. We also formally tested whether genetic correlations are not significantly lower than 1 after

Bonferroni p-value adjustment. We applied the same statistical testing framework as used by the linkage disequilibrium score regression (LDSC) software for assessing whether genetic correlations between FTND and another phenotype are statistically significant (Pearson's chi square test). These results were added to Supplementary Table 11.

3. *Did the authors consider to add PrediXcan analyses to test whether imputation of gene expression identifies additional loci beyond single SNP eQTL analysis? I was actually quite surprised that they did not conduct any type of gene-based testing (e.g., MAGMA).*

Response: We appreciate this suggestion. In response, we have added results from two gene-based association methods using our GWAS summary statistics from iNDiGO as the input: Summary-MultiXcan (S-MultiXcan)⁹ (an extension of PrediXcan) that incorporates *cis*-eQTL evidence across multiple tissues (specifically, 13 brain tissues with RNA-sequencing in GTEx as the reference data) and Hi-C coupled multi-marker analysis of genomic annotation (H-MAGMA)¹⁰ that takes into account physical proximity (as done in MAGMA) as well as chromatin interaction maps from both fetal and adult brain tissues in annotating SNPs to derive gene-level association results. Results from these gene-level methods corroborate associations for the previously identified loci, particularly in chromosome 15q25 and 20q13. These new analyses are now outlined at lines 412–434 (Methods), 125–144 (Results), and 235–237 and 266–268 (Discussion) and in the new Supplementary Table 9 for H-MAGMA. The corresponding genome-wide results are provided in the new Supplementary Tables 7–8 for H-MAGMA and Supplementary Table 10 for S-MultiXcan.

4. *Page 9, paragraph on gene expression. Can the authors provide more information on which resources have been used?*

Response: We expanded our description of the underlying gene expression data that were used to conduct our stratified LDSC, as applied to specifically expressed genes (LDSC-SEG), analyses in both the Methods (lines 446–449) and Results (lines 215–218; per the reviewer's request) sections.

5. *The authors could do more to describe the genetic overlap in terms of pleiotropy, e.g., pairwise GWAS (Pickrell et al) or MiXeR (Frei et al) to identify loci that are unique for a specific trait vs. loci that are shared across traits.*

Response: We attempted application of both MiXeR¹¹ and pairwise GWAS¹² to compare GWAS results from ND and its correlated traits from our original LDSC analyses. Both methods use GWAS summary statistics as inputs to quantify overlap in genetic variant associations that are shared between two traits vs. trait-specific variant associations. Because both methods harness LD information, like LDSC, we needed to rely on the European ancestry-specific GWAS meta-analysis results (N=46,213). However, unlike the correlation-based LDSC method, the performance of both MiXeR and pairwise GWAS are dependent on having large sample sizes for the datasets being compared, as stated in the publication¹² or communication with the method developer (Dr. Frei, first author of MiXeR¹¹), and our ND GWAS results were underpowered for these genome-wide analyses. However, we were able to perform pairwise GWAS in a targeted fashion, to assess the pleiotropic evidence for our 5 genome-wide significant loci. Results showed that our novel locus on chr5 (rs1862416) had shared genetic influence with major depressive disorder and smoking initiation, but our chr7 locus (rs2714700) did not produce a high probability of containing a variant that influences both ND and the other traits. However, this locus did show evidence for variants influencing alcohol dependence, depressive symptoms, and

schizophrenia that are distinct from variants influencing ND. These results have been added to the main body of the manuscript (lines 163–175, 319–322, and 460–474) and included as a new supplementary figure (Supplementary Figure 5).

Lastly, in our deeper investigation into the pleiotropic associations of our genome-wide significant loci, we found a recently published *TENM2* variant (rs2337033) association with cigarette pack-years¹³, located near our ND-associated rs1862416 but weakly correlated (r^2 of 0.22 in 1000 Genomes EUR and 0.11 in AFR reference populations). We have revised Supplementary Figure 6 and Discussion (line 280) accordingly with this new published finding, which adds even more evidence for pleiotropy at the gene level for *TENM2*. With our implication for ND, *TENM2* variants have now been identified at genome-wide significance for at least 15 complex traits, including addiction and psychiatric traits.

6. Genomic SEM would be very useful to explore the genetic factor structure of the FTND items. I think additional analyses will be important since the number of novel loci is relatively limited.

Response: We agree that genomic SEM is a well-suited method that may help to compare and contrast the genetic architecture underlying the 6 FTND times. However, we feel that this new analysis is beyond the scope of the current paper, amounting to 138 new GWAS analyses (GWAS for each of 6 FTND items × 23 cohorts), in the timeframe allotted. The volume of results afforded by genomic SEM could serve as the basis for future study, as we now acknowledge in the Discussion (lines 346–348).

Reviewer #2's Comments

7. Functional annotation, page 8-9, the authors concentrate their functional annotation to the two index variants in the novel genome-wide significant loci. However, the causal variant is not necessarily the index variant of a locus but could be any variant in high LD with the index variant. The authors should perform functional annotation of all variants/or a set of credible variants in the two regions and preferably use a fine mapping method that takes ancestry into account (e.g the method by Lam et al. Nature Genetics 2019).

Response: We agree the index SNP may not be the true causal variant underlying a genetic association signal and that the functional annotations should be expanded. As suggested, we have now updated our manuscript to include a credible set analysis (see Methods, lines 476–478) and expanded the annotation to include variants beyond the index SNP, shown in the new Supplementary Table 12 and presented in the Results (lines 191–211). The fine mapping method used by Lam et al. 2019¹⁴ is an important and valid approach to identify potentially causal variants by comparing signals across different populations based on the assumption that causal variants will have more statistical significance and decreased heterogeneity. However, because we observed lower P-value signals in the African Americans and very little heterogeneity in our cross-ancestry meta-analysis due to comparable effect sizes, we do not expect this method would identify additional variants. We confirmed this by discussions with Drs. Lam and Huang (first and corresponding author of Lam et al. Nature Genetics 2019) who verified the method would not provide additional information for our particular dataset. To clarify this issue, we have provided the ancestry-specific meta-analysis results and heterogeneity p-values for the credible set in Supplementary Table S12.

8. The authors should provide more information about the phenotype tested already in the beginning of the result section. The authors write in the introduction that FTND provides information about a

composite phenotype. It should therefore be clear already from the start of the result section (not only the methods section) how the authors have defined the phenotype used in the GWAS meta-analysis. Additionally, it is confusing how they have defined ND based on the FTND scores. In the result section page 5, bottom, the authors write that they have tested severe vs. mild ND, but in the method section page 16, they write that they have tested for association with a 3-level categorical ND. Please clarify this issue.

Response: In the Introduction (lines 45–53, the paragraph immediately preceding the Results section), we have expanded the description of the FTND and specified how we categorized its score (ranging from 0 to 10) into a 3-level categorical phenotype (mild, moderate, and severe ND) for our prior and current GWAS analyses. Further, we have clarified at first occurrence, in the Results section rather than the Methods, that the severe vs. mild results were derived directly from the GWAS linear regression model coefficients to contextualize the magnitude of our observed effect sizes (lines 80–83).

9. The authors test for genetic correlation with 45 complex phenotypes. For some of the phenotypes analyzed the authors refer to papers with outdated data in Supplementary Table 3. It is therefore not clear if the data they have used should be updated or the references are wrong. I have not gone through all references but can see that the references to ADHD is outdated (should instead be Demontis et al. 2019), Autism spectrum disorders (should instead be Grove et al. 2019), and bipolar disorder is outdated as well. Please clarify these analyses.

Response: We thank the reviewer for careful attention paid to these results. Our original LD score regression analyses relied heavily on summary statistics that are deposited in LD Hub. We revisited the analyses and obtained more up-to-date GWAS summary statistics (made available outside of LD Hub) to enable updated genetic correlations for the following phenotypes: anorexia nervosa, attention deficit hyperactivity disorder, autism spectrum disorder, bipolar disorder, major depressive disorder, posttraumatic stress disorder, years of schooling, and lung cancer (overall and histological subtypes [adenocarcinoma, small cell, and squamous]). Figure 3, Supplementary Table 11 (formerly labeled Supplementary Table 3, as referenced by the reviewer), and the main text (Abstract lines 12–13, Methods line 438 and 458, Results lines 145–162, and Discussion lines 319–322) have all been updated accordingly. With the updated results, we now report 17 phenotypes, rather than 13 phenotypes from our original submission, that have significant genetic correlations with nicotine dependence.

10. Page 5: “AA-specific” has not previously been defined.

Response: “AA” refers to African American, as indicated on page 6 (line 68), but to clarify, we have revised this sentence to read “No genome-wide significant loci were found in the GWAS meta-analysis among AAs ($\lambda=1.032$, Supplementary Figures 1C and 2B).”

11. Page 6: Please clarify if there are any sample overlap between GSCAN and iNDiGO samples.

Response: iNDiGO and GSCAN are not mutually exclusive, and we have now specified that European ancestry participants from 8 of the iNDiGO studies were included in GSCAN (lines 91–92).

12. Page 6 bottom: I would prefer if you replace “For independent testing” with “For replication in an independent sample”.

Response: We have revised this sentence, as suggested (line 96).

13. *Page 6, bottom: Please already here describe the difference between HSI and FTND. For someone not familiar with these measures it would be nice know at this point that HSI comprise TTFC and CPD.*

Response: We have revised this Results paragraph to describe the heaviness of smoking index (HSI), which is comprised of two items (cigarettes per day and time-to-first cigarettes) of the 6 FTND items, and to present the high correlation (up to $r=0.9$) between HSI and the full-scale FTND (lines 98–100).

14. *Page 7, top: for consistency replace “all discovery samples” with “iNDiGO samples”.*

Response: We have replaced “all discovery” with “iNDiGO” as suggested.

15. *Page 7, second paragraph: “...the factors of ND”. This wording is unspecific, please specify what is meant by factors.*

Response: We have revised the original wording “To determine the factors of ND that drove the novel genome-wide significant associations ...” to now read “To determine whether the novel genome-wide associations were driven by specific FTND items or shared across items” (lines 105–106).

16. *Page 7, second paragraph: “TTFC” is mentioned for the first time. Specify what it means.*

Response: Thank you for pointing out this oversight. We have spelled out the abbreviation at the first mention of TTFC (line 40).

17. *Page 9, second paragraph: “To assess the enrichment of the EUR-specific ND...” I assume it is enrichment in the heritability, but this is not clear.*

Response: The reviewer is correct, and we have clarified the intent of the LDSC-SEG analyses (i.e., to assess whether heritability of ND is enriched in regions surrounding genes with the highest specific gene expression patterns in given tissue/cell types) at lines 212–213.

18. *Page 10, second paragraph: The authors mention “primary and secondary features of ND”. This is defined later in the discussion, but it would help to have this information already at this point.*

Response: We have expanded this paragraph to present the concepts of primary and secondary features of ND earlier in the Discussion section (lines 243–246).

19. *Page 11 top: rs4307059 is not genome-wide significantly associated with autism.*

Response: The originally cited reference focused on the potential functional implications for rs4307059. With this comment, we have now added the citation for the autism GWAS from which this SNP was identified at genome-wide significance and independently replicated¹⁵. See line 259.

20. *Figure 2, item 3: “Cigarette most hated to give up”, this description does not provide sufficient information about what the variant is associated with.*

Response: We have updated this Figure 2 notations for item 3 to “Cigarette most hated to give up (first in the morning or others)” as well as item 1 to “Time to first cigarette after waking”.

21. Page 1, second paragraph: "...yet despite the nearly complete sharing between ND and CPD.." this sentence is not clear. By "nearly complete sharing" I assume the authors mean a genetic correlation close to one, but this is not clear, please rephrase.

Response: The reviewer's assumption is correct, and we have revised the sentence for clarity: "We observed statistically significant genetic correlations of each of these smoking traits with ND (highest $r_g=0.95$, as observed between ND and CPD), yet our two novel ND-associated loci were not identified by GSCAN ..." (lines 293–294).

22. Page 12, second paragraph: "These observations resemble previously reported patterns of genetic correlation between alcohol dependence and alcohol consumption". I do not agree with this statement. First of all, alcohol consumption and alcohol dependence disorder demonstrate opposite genetic correlations with several traits including educational attainment where the former is positively correlated and the latter negatively. Both CPD and ND demonstrate negative genetic correlation with educational attainment. Additionally, just because some loci are ND specific compared to CPD, it does not support the existence of similar pattern as the one observed at the polygenic level for alcohol dependence vs alcohol use disorder.

Response: We have removed this sentence.

23. Page 13, first paragraph: what is Taq1A – if it is a gene, it should be in italics.

Response: 'Taq1A polymorphism' refers to the historical name for the *DRD2* variant rs1800497. We have revised this sentence (line 307) to clarify that rs1800497 and Taq1A polymorphism are interchangeable names for the same genetic variant.

24. Page 13, third paragraph: insert references for the disorders included in the genetic correlation analyses.

Response: We have added the references, including those updated in response to comment #10, at lines 319–322.

25. Page 16, third paragraph: Please provide information about filtering on imputation info score, and have variants in the GWAS meta-analysis been filtered based on the effective sample size?

Response: SNPs/indels were filtered based on minor allele frequency < 1%, imputation info score < 0.3, or availability in a single study, as we have clarified at line 384–385. No additional filtering was performed based on effective sample size.

References

1. Hancock, D.B. *et al.* Genome-wide association study across European and African American ancestries identifies a SNP in DNMT3B contributing to nicotine dependence. *Mol Psychiatry* **23**, 1911-1919 (2018).
2. Breslau, N. & Johnson, E.O. Predicting smoking cessation and major depression in nicotine-dependent smokers. *Am J Public Health* **90**, 1122-7 (2000).
3. Agrawal, A. *et al.* A latent class analysis of DSM-IV and Fagerstrom (FTND) criteria for nicotine dependence. *Nicotine Tob Res* **13**, 972-81 (2011).

4. Baker, T.B. *et al.* Are tobacco dependence and withdrawal related amongst heavy smokers? Relevance to conceptualizations of dependence. *J Abnorm Psychol* **121**, 909-21 (2012).
5. Baker, T.B. *et al.* Time to first cigarette in the morning as an index of ability to quit smoking: implications for nicotine dependence. *Nicotine Tob Res* **9 Suppl 4**, S555-70 (2007).
6. Sweitzer, M.M., Denlinger, R.L. & Donny, E.C. Dependence and withdrawal-induced craving predict abstinence in an incentive-based model of smoking relapse. *Nicotine Tob Res* **15**, 36-43 (2013).
7. Bolt, D.M. *et al.* The Wisconsin Predicting Patients' Relapse questionnaire. *Nicotine Tob Res* **11**, 481-92 (2009).
8. Haberstick, B.C. *et al.* Genes, time to first cigarette and nicotine dependence in a general population sample of young adults. *Addiction* **102**, 655-65 (2007).
9. Barbeira, A.N. *et al.* Integrating predicted transcriptome from multiple tissues improves association detection. *PLoS Genet* **15**, e1007889 (2019).
10. Sey, N.Y.A. *et al.* A computational tool (H-MAGMA) for improved prediction of brain-disorder risk genes by incorporating brain chromatin interaction profiles. *Nat Neurosci* (2020).
11. Frei, O. *et al.* Bivariate causal mixture model quantifies polygenic overlap between complex traits beyond genetic correlation. *Nat Commun* **10**, 2417 (2019).
12. Pickrell, J.K. *et al.* Detection and interpretation of shared genetic influences on 42 human traits. *Nat Genet* **48**, 709-17 (2016).
13. Buchwald, J. *et al.* Genome-wide association meta-analysis of nicotine metabolism and cigarette consumption measures in smokers of European descent. *Mol Psychiatry* (2020).
14. Lam, M. *et al.* Comparative genetic architectures of schizophrenia in East Asian and European populations. *Nat Genet* **51**, 1670-1678 (2019).
15. Wang, K. *et al.* Common genetic variants on 5p14.1 associate with autism spectrum disorders. *Nature* **459**, 528-33 (2009).

REVIEWERS' COMMENTS:

Reviewer #1 (Remarks to the Author):

The authors have done an excellent job revising this manuscript. I have no further suggestions for improvement.

Reviewer #3 (Remarks to the Author):

The authors have made extensive efforts to address the initial concerns of the reviewers, and this has led to substantial improvements to the manuscript. I have just two additional comments.

1.) Given the overlap in items and findings regarding the FTND phenotype used in the discovery analysis and the "heaviness of smoking" phenotype used in the replication analysis, it would be helpful to know, at minimum, the genetic correlation between these phenotypes. The authors could also choose to include results from or simply comment on whether it would be appropriate to include results from a combined analysis.

2.) The described gene-based tests and enrichment analyses point to different brain regions (e.g., substantia nigra vs. hippocampus), but there is no explanation or interpretation of these discrepant results. It is not unusual to encounter such discrepancies given the imperfect nature of annotation data and the multitude of methods that leverage it, but the large number of presented results and potential lack of familiarity of some readers with these methods could lead to confusion. Thus, some comment on how readers should attempt to reconcile these findings would be helpful.

We appreciate the reviewers' thoughtful comments. We believe that addressing their comments (shown in italics), as detailed point-by-point below, has strengthened our study's results and conclusions.

Reviewer #1's Comment

The authors have done an excellent job revising this manuscript. I have no further suggestions for improvement.

Response: Thank you for this compliment!

Reviewer #3's Comments

The authors have made extensive efforts to address the initial concerns of the reviewers, and this has led to substantial improvements to the manuscript. I have just two additional comments.

Response: We appreciate the compliment, and address the two additional comments below.

- 1. Given the overlap in items and findings regarding the FTND phenotype used in the discovery analysis and the "heaviness of smoking" phenotype used in the replication analysis, it would be helpful to know, at minimum, the genetic correlation between these phenotypes. The authors could also choose to include results from or simply comment on whether it would be appropriate to include results from a combined analysis.*

Response: We have computed the genetic correlation between the FTND (based on the iNDiGO EUR-specific GWAS meta-analysis) and the heaviness of smoking index (HSI, based on our analysis of the UK Biobank), resulting in r_g (standard error) = 1.09 (0.15). Of note, the LD score regression estimate for genetic correlation (r_g) is not bounded by [-1, 1]. Given this very strong genetic correlation between the FTND and HSI phenotypes, we completed a new GWAS meta-analysis combining the FTND-based iNDiGO cohorts with the HSI-based UK Biobank and present results in the newly added Supplementary Figures 7–8 and Supplementary Table 9. This new meta-analysis identifies 7 genome-wide significant loci, including one locus surrounding the *ARHGAP25* gene on chromosome 2 that has not been previously reported for any smoking trait. We discuss these new results on lines 227–239 of the Results and lines 305–309 of the Discussion.

- 2. The described gene-based tests and enrichment analyses point to different brain regions (e.g., substantia nigra vs. hippocampus), but there is no explanation or interpretation of these discrepant results. It is not unusual to encounter such discrepancies given the imperfect nature of annotation data and the multitude of methods that leverage it, but the large number of presented results and potential lack of familiarity of some readers with these methods could lead to confusion. Thus, some comment on how readers should attempt to reconcile these findings would be helpful.*

Response: The reviewer raises an important point, and we have added text to the Discussion section (lines 362-375) to discuss possible reasons for seemingly discrepant findings.